# METASPACE-ML: Context-specific metabolite annotation for imaging mass spectrometry using machine learning

Bishoy Wadie[1,2,8], Lachlan Stuart [1,8], Christopher M. Rath[1], Bernhard Drotleff[3], Sergii Mamedov [1] & Theodore Alexandrov [1,3,4,5,6,7] ✉

Imaging mass spectrometry is a powerful technology enabling spatial metabolomics, yet metabolites can be assigned only to a fraction of the data generated. METASPACE-ML is a machine learning-based approach addressing this challenge which incorporates new scores and computationally-efficient False Discovery Rate estimation. For training and evaluation, we use a comprehensive set of 1710 datasets from 159 researchers from 47 labs encompassing both animal and plant-based datasets representing multiple spatial metabolomics contexts derived from the METASPACE knowledge base. Here we show that, METASPACE-ML outperforms its rule-based predecessor, exhibiting higher precision, increased throughput, and enhanced capability in identifying low-intensity and biologically-relevant metabolites.

Imaging mass spectrometry (imaging MS) has emerged as a leading technology in spatial metabolomics, finding applications in diverse fields such as biology, medicine, and pharmacology[1–3]. However, a key challenge remained in the accurate and confident annotation of metabolites, primarily due to limitations in data collection, existing algorithms, and software[4]. Similar to bulk metabolomics[5], the vast majority of the imaging MS data, so-called "dark matter", cannot be molecularly annotated with existing tools.

We previously developed METASPACE, an engine for metabolite annotation[6] and a community-populated knowledge base[7]. METASPACE utilizes a False Discovery Rate (FDR)-controlled approach, where metabolite ions are reported at a given confidence level by ranking them against implausible generated decoy ions. However, a limitation of METASPACE is its rule-based scoring system, namely Metabolite Signal Matching (MSM), which assigns equal weights to the features and lacks adaptability to data variations. Previous attempts to employ a data-driven rescoring approach did not consistently increase the number of annotations across diverse datasets[8].

Recently, METASPACE evolved into the leading repository for spatial metabolomics, acting as an extensive knowledge base with over 10.000 public datasets contributed by researchers globally. This opened an opportunity to train a machine-learning model on this big data.

Here, we introduce METASPACE-ML, a machine-learning model for FDR-controlled metabolite annotation within specific contexts in imaging MS, incorporating new features developed for centroid data. We devise a strategy to select training and testing datasets representing contexts of different technologies and sample types. METASPACE-ML is trained on 600 animal datasets and 180 plant datasets, and its performance is evaluated on 930 datasets submitted by 47 different labs. We demonstrate that METASPACE-ML outperforms the traditional MSM rule-based approach by delivering more annotations, especially at low FDR thresholds and for ions of low intensity. We also investigate the enrichment of molecular classes in datasets with the most significant improvement and compare predicted annotations against an untargeted LC-MS/MS analysis. Finally, we offer an efficient implementation using the Lithops cloud serverless computing framework[9] and integrate METASPACE-ML into the existing cloud METASPACE software as well as a separate web app (https://t.ly/q-nb5) to help users align

[1]Structural and Computational Biology Unit, European Molecular Biology Laboratory (EMBL), Heidelberg, Germany. [2]Collaboration for joint PhD degree between EMBL and Heidelberg University, Faculty of Biosciences, Heidelberg, Germany. [3]Metabolomics Core Facility, EMBL, Heidelberg, Germany. [4]Molecular Medicine Partnership Unit, EMBL, Heidelberg, Germany. [5]BioStudio, BioInnovation Institute, Copenhagen, Denmark. [6]Department of Pharmacology, University of California San Diego, La Jolla, CA, USA. [7]Department of Bioengineering, University of California San Diego, La Jolla, CA, USA. [8]These authors contributed equally: Bishoy Wadie, Lachlan Stuart. ✉e-mail: talexandrov@health.ucsd.edu

their datasets to the defined context and check how well the model performs in such contexts.

## Results

### Key principles of METASPACE-ML

We aimed to optimize the discrimination between target ions and implausible decoy ions[6] by employing a data-adaptive scoring method for FDR-controlled annotation (Fig. 1A). We used five scores per ion, including three constituents from the MSM score[6], and two additional scores estimating the absolute and relative mass-to-charge (m/z) error (Fig. 1B).

Target and decoy ions were separated with a ranking-based Gradient Boosting Decision Trees (GBDT)[10]. An ensemble of decision trees was trained iteratively, with each subsequent tree correcting the errors of the previous ones. During each iteration, the target-decoy ion pairs from the training data were scored using a decision tree, aiming to minimize the PairLogit loss function that reflects the difference in their prediction scores (Fig. 1C). Training the ensemble produced the METASPACE-ML model (Fig. 1D) which can be used to score each target ion with a continuous FDR score. In addition, it provides users with comprehensive information regarding the contribution and impact of each feature on the final prediction score. A brief summary of training and evaluation of the model is shown in Fig. 1E.

### Representative selection of datasets

METASPACE hosts the largest public collection of spatial metabolomics datasets spanning different organisms, sample types, and imaging mass spectrometry (imaging MS) technologies and protocols. Despite the sheer amount of public datasets (over 10.000 as of January 2024), technologies and different sample types are not equally represented, which is to be expected because METASPACE is populated by submissions from the imaging MS community. Out of 9251 public datasets downloaded from METASPACE in September 2023, 7713 datasets passed our quality criteria, 84% and 14.3% of which correspond to animal and plant datasets, respectively (Fig. 2A). Of them, 72% of the datasets were acquired with MALDI-imaging MS, 47% of datasets are from human samples, ~ 60% are from tissue sections, and Orbitrap datasets represent 29% and 62% of plant and animal datasets, respectively (Fig. 2B and Supplementary Fig. 1). We chose six different categories (hereafter referred as contexts) to classify public datasets (see "Methods") to reflect the diversity of submissions in METASPACE. To make sure that all the contexts are equally represented in model training, we kept the number of datasets per context fixed (see "Methods"). In addition, we performed training and testing for animal and plant datasets independently. We also tested different context sizes per kingdom (Supplementary Fig. 2), and since the difference between the training and validation error for context size ≥ 30 are minimal (Supplementary Fig. 3) and to ensure that more contexts are represented, we chose the model with 30 datasets per context going forward. Accordingly, the training set comprised 600 and 180 animal and plant datasets, respectively (Fig. 2C and Supplementary Fig. 4). The testing set comprised 720 and 210 animal and plant-based datasets, respectively (Fig. 2D, Supplementary Fig. 5). In total, 1710 datasets were selected for training and testing of the new METASPACE-ML model. For animal-based training datasets, we had an almost balanced distribution of positive and negative polarity, MALDI was the most prevalent ionization source (510 out of 600), often coupled with Orbitrap or FTICR analyzers. Approximately 75% of the datasets represented samples from either human or mouse. Cell-based datasets were exclusively from human-derived samples (Fig. 2C). For plant-based training datasets, MALDI was the only ionization source represented, often coupled with the FTICR analyzer unlike animal-based datasets where Orbitrap analyzers were prevalent (Fig. 2C and Supplementary Fig. 4). Approximately 67% represented common crops (*Populus* and *Sorghum*) and most of the datasets are tissue-based

(Supplementary Fig. 4). In summary, the context-specific selection of datasets helped us select a large representative set of datasets and will provide the end-users the granularity of the represented sample types and protocols to judge on the reliability of the model predictions.

### METASPACE-ML outperforms the rule-based approach in ranking quality

To assess the performance of the METASPACE-ML model, we conducted a comparative evaluation against the state-of-the-art rule-based MSM approach[6]. The mean average precision (MAP) was used to evaluate the quality of target vs decoy ion rankings.

Overall, METASPACE-ML achieved higher median MAP scores compared to the rule-based approach (0.36 vs 0.27) and (0.32 vs 0.17) on both animal and plant cross-validated datasets, respectively (Supplementary Fig. 6A, B). Similarly, METASPACE-ML had significantly ($p$-value $= 2 \times 10^{-16}$) higher median AP scores in 674 out of 720 and 207 out of 210 animal and plant-based testing datasets, respectively (Fig. 3A and Supplementary Fig. 7).

Breaking down ranking quality by context, we observed that METASPACE-ML outperforms the rule-based approach in all contexts, with noticeable differences in MAP scores in tissue and MALDI-Orbitrap contexts in animal-based datasets (Supplementary Fig. 8) and contexts association with *Populus* in plant-based datasets (Supplementary Fig. 9). Moreover, using only the METASPACE-ML score, human and tissue-based MALDI Orbitrap in negative mode and *Populus* and tissue-based MALDI-FTICR in negative mode had the highest median AP compared to other contexts in animal and plant-based datasets, respectively (Fig. 3B and Supplementary Fig. 10). Significance of pairwise context comparison of MAP scores using the Wilcoxon test can be found in Supplementary Figs. 11, 12.

Furthermore, when using different annotation databases, we observed that METASPACE-ML still outperforms the rule-based approach in overall MAP, with particularly promising results for the CoreMetabolome database (Fig. 3C and Supplementary Fig. 13). This is in line with the AP distributions, where using CoreMetabolome led to a significant (Friedman $p$-value < 0.001) median AP compared to other databases (Fig. 3D).

### METASPACE-ML captures more highly-confident annotations

In addition to the overall ranking quality, we considered the number of low-FDR annotations as a measure focused on results of high confidence. Accordingly, we calculated either difference or fold change between the numbers of annotations to compare how better is the METASPACE-ML model relative to its rule-based predecessor.

Overall, at the default FDR 10%, METASPACE-ML, identified approximately 20 and 70 more annotations on average and at least 50 and 100 more annotations in 50% of the animal and plant-based testing datasets, respectively (Fig. 4 A, B). At FDR 5%, METASPACE-ML revealed approximately a 1.64-fold and 1.80-fold increase in annotations compared to the rule-based approach in 50% of the animal and plant-based datasets (Supplementary Figs. 14, 15), indicating a significant ($p$-value < 0.01) improvement for most-confident annotations at lower FDR.

Breaking down annotation coverage by context, we observed that METASPACE-ML, on average, captures more annotations than rule-based approach in all contexts for both animal and plant-based datasets at 10% FDR (Fig. 4C and Supplementary Fig. 16). However, Orbitrap based contexts had a significantly ($p$-value $= 7.93 \times 10^{-20}$) higher Log10 difference in animal-based datasets compared to FTICR-based contexts with a median of 70 more annotations (Fig. 4D). Interestingly, single-cell based datasets had a comparably high Log10 difference which is comparable with the Orbitrap tissue based contexts (Fig. 4C). In plant-based datasets, FTICR and *Populus* datasets in negative polarity had the highest Log10 difference with an average of 200 more annotations (Supplementary Fig. 16). Furthermore, if we compare the

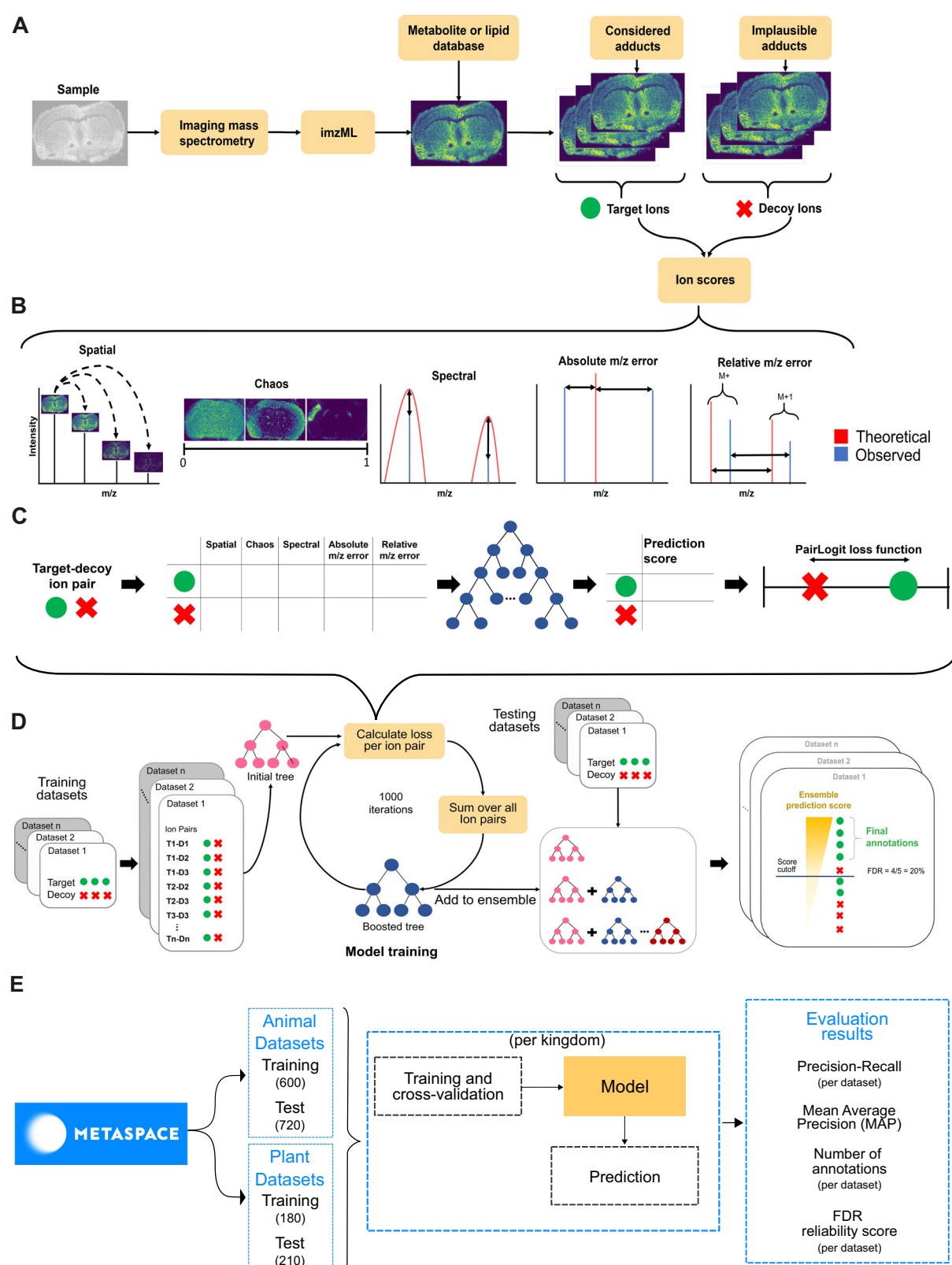

**Fig. 1 | Workflow demonstrating key aspects of METASPACE-ML, model training, and evaluation. A** The workflow from the user perspective covers steps of data acquisition, molecular database selection, and preparation of the target and decoy database. **B** Scores are calculated for each ion which are used as features for the machine learning model. **C** The principle of machine learning is applied to the target and decoy ions' features which are scored by a single decision tree by maximizing the PairLogit loss function. **D** Details of the training of Gradient Boosting Decision Trees using multiple datasets and visualization of how the metabolite annotations are generated for a given FDR threshold. **E** Details on the public datasets from METASPACE used for training and evaluation, as well as the measures used for evaluation. Created with Inkscape v1.2 and yED graph editor v3.22.

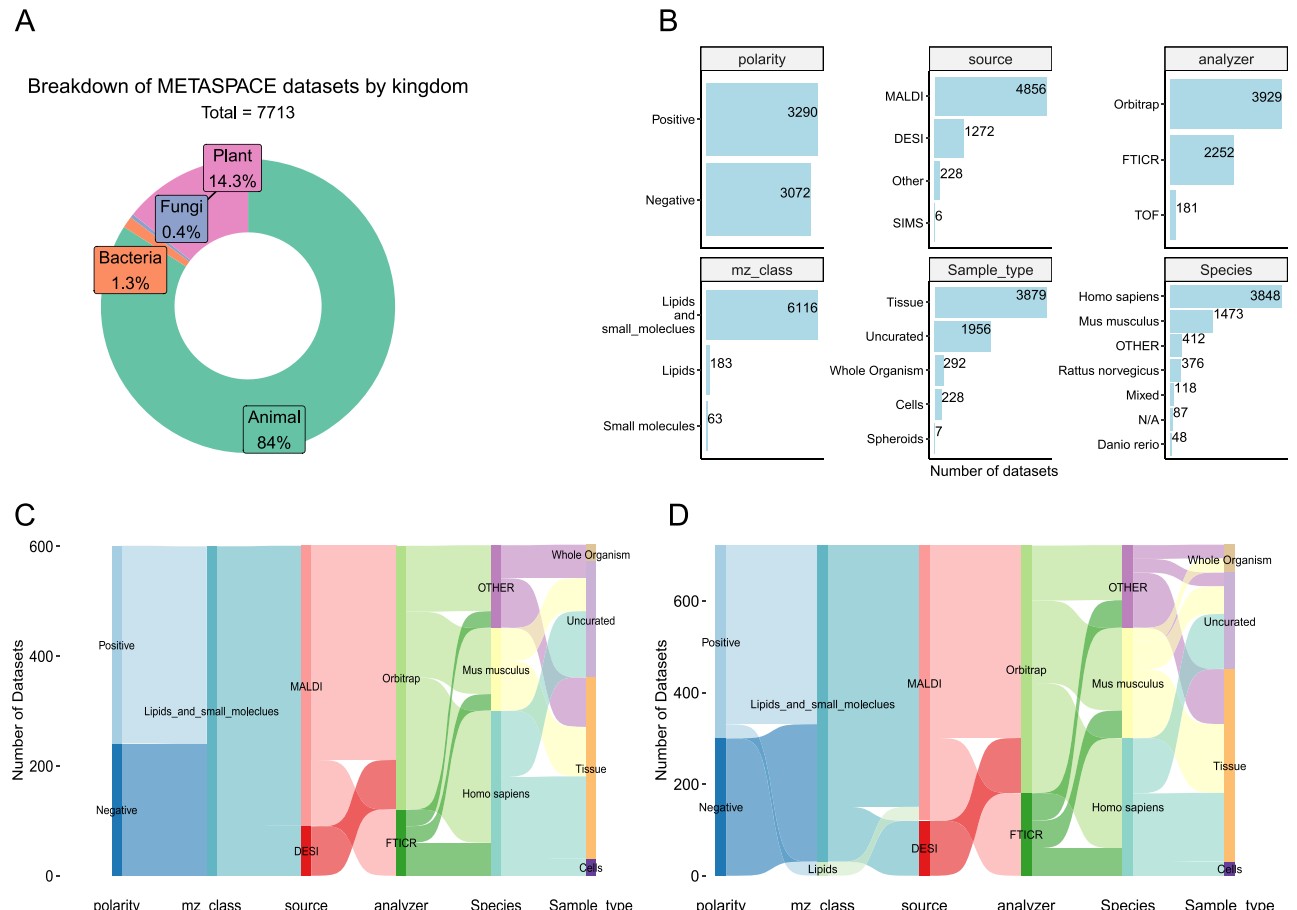

**Fig. 2 | Public METASPACE datasets used for training and evaluation and details on representation of different technological and biological contexts.**
**A** Doughnut plot showing a breakdown of public METASPACE datasets selected for training and evaluation after applying the quality filters. Each sector is colored by the kingdom and depicts the percentage of the total datasets for each kingdom.
**B** Breakdown of the animal datasets by their associated technological and biological metadata making up the contexts (see methods). A number of datasets are shown on the *x*-axis, and the classes in each context are shown on the *y*-axis. **C**, **D** Sankey diagrams show the breakdown of training (**C**) and testing (**D**) animal datasets by different parameters. Properties of the datasets are described on the *x*-axis and the corresponding numbers of datasets in each level are plotted on the *y*-axis. Each property can be divided into multiple classes, which are represented by the colors of the nodes and their corresponding flow to the next layer. A flow from the first to the last node represents a single technology-biology context.

median log-fold change (LFC) across different FDR thresholds per context, we still observe the highest LFC at FDR 5% in almost all animal-based contexts, except MALDI-Orbitrap in negative polarity for both human and mouse uncurated datasets as well as MALDI-FTICR in negative polarity for mouse tissue datasets (Supplementary Fig. 17). In plant-based contexts, the LFC distribution per context across different FDR thresholds is more variable with FDR 5% not necessarily having the highest median LFC (Supplementary Fig. 18).

Finally, comparing different annotation databases at the default 10% FDR, we observed that METASPACE-ML, on average, identifies slightly more annotations using LipidMaps and SwissLipids compared to CoreMetabolome (Fig. 4E). We also observed a greater improvement in LFC for most-confident annotations at lower FDR (especially at 5 % FDR). This trend was consistent for all databases, with higher relative improvements for LipidMaps and SwissLipids at lower FDRs compared to CoreMetabolome (Supplementary Fig. 19).

**METASPACE-ML detects more annotations with a higher reliability**
Assessing and ensuring the reliability of METASPACE-ML is pivotal to informing decisions and improving the explainability of the predictions. First, building on the aforementioned performance metrics (MAP and difference in the number of annotations), it's essential to

understand the relationship between the annotation coverage and the ranking quality for each dataset in order to identify patterns and discrepancies where marginal effects can take place (e.g., more low-FDR annotations with low MAP). By plotting the Log10 difference against MAP, we found that METASPACE-ML captures 30–1000 more annotations than the rule-based approach with MAP > 0.2 in around 55% and 65 % of animal-based and plant-based datasets at 10% FDR (Fig. 5A and Supplementary Fig. 20). Grouping datasets into whether the ML models captured more, less or equal numbers of annotations compared to its rule-based predecessor, we found that datasets, where the ML model captured less annotations, had significantly lower (*p*-value < 0.001) MAP values compared to the other categories in both animal and plant-based datasets (Fig. 5B and Supplementary Fig. 21).

Given the inherent imbalance in the number of target vs decoy ions for each dataset (as we randomly sample *n* = 20 decoy ions for each target ion), there are more negative cases than there are positive cases to classify. Therefore, one might be concerned about the precision of the model and its false positive rate (FPR) as opposed to the recall. This concern is particularly relevant when evaluating the discriminatory ability of the FDR scores for each annotation, especially under various FDR thresholds. Therefore, we formulated a reliability score (*reliability_score*) for the results of metabolite annotation using a target-decoy approach. The score ranges from 0 (low reliability) to 1

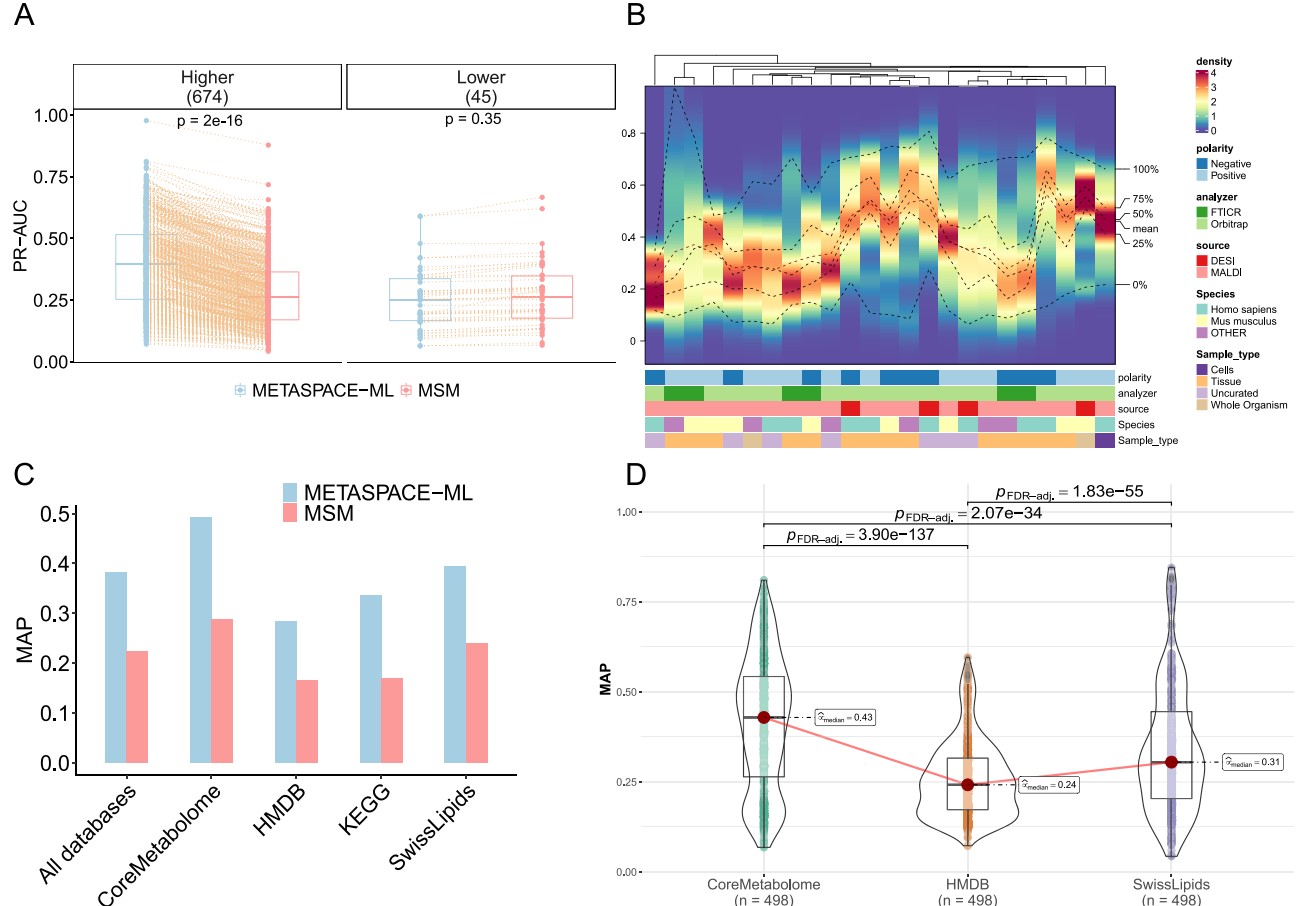

**Fig. 3 | Assessment of ion ranking quality for METASPACE-ML vs its rule-based predecessor. A** Paired boxplot showing Area Under precision-recall Curve (AUC) per dataset for animal testing datasets. Approach type and AUC scores are shown on the *x*-axis and *y*-axis, respectively. Each dot represents a dataset and an orange dotted edge is drawn between the same datasets annotated by both approaches. Boxplots are grouped by the difference of AUC scores (Delta) where positive differences denote higher AUC in METASPACE-ML relative to MSM and vice-versa. The number of datasets in each group is shown in parentheses. Exact *p*-values are based on a two-tailed paired Wilcoxon signed-rank test between AUC scores across both approaches. **B** Density heatmap showing MAP score distributions across datasets for each context in the animal testing datasets. Each column represents a context, described by its metadata, with colors representing different classes. The *y*-axis shows MAP scores, and the color gradient represents density. Columns are hierarchically clustered using the Kolmogorov-Smirnov statistic, with dotted lines

indicating different quantiles and the mean. **C** Bar graph showing MAP scores for ($n = 389$) testing datasets annotated against four databases separately and all databases combined. AP (average precision) scores are calculated for each group (dataset and adduct) and the score is the average across all groups for a given database. **D** Box-violin plot displaying the distribution of MAP scores across ($n = 389$) animal testing datasets, compared across different annotation databases using a two-tailed paired Friedman test with *p*-values adjusted by the Benjamini & Hochberg method. MAP scores and annotation databases are shown on the *x*-axis and *y*-axis, respectively. Adjusted *p*-values between pairwise comparisons of databases are shown between each comparison. Only significant (adjusted *p*-value < 0.05) comparisons are shown. In (**A**, **D**), the boxplots' bottom and top edges represent the 25th and 75th percentiles, with the median (50th percentile) line inside the box. Whiskers are omitted; minimum and maximum values are represented by jittered data points.

(high reliability) and employs the diagnostic F-beta measure (with beta = 0.5). We propose to use the *reliability_score* to estimate the reliability of the annotation results and choose the most reliable FDR threshold for each dataset (the minimal FDR that maximizes the score). After considering optimal FDR thresholds for each dataset, we observed a significant (*p*-value < 0.001) improvement in LFC for the numbers of annotations compared to all other fixed FDR thresholds (Fig. 5C and Supplementary Fig. 22) and we found that 70% of animal datasets had optimal FDR thresholds at 5% with a median reliability score of 0.88 compared to the other default thresholds (Fig. 5D). Plotting Log10 difference against MAP at custom thresholds, we observed that METASPACE-ML captured 30–1000 more annotations than rule-based approach with MAP > 0.2 in around 64% and 65% of animal and plant-based datasets (Supplementary Figs. 23, 24). Our reliability score *reliability_score* will be a useful metric for the end-users to evaluate their results and identify the optimal FDR threshold that

optimizes both precision and recall, with a higher weight on the former.

**Assessment of target-decoy separation and feature importance**
After examining the performance and reliability of METASPACE-ML predictions, it's essential to understand how the features constituting the model impact the prediction score and the separation of targets from decoy ions which is crucial for FDR-controlled annotation. Using SHAP values (SHapely Additive exPlanations) we observed that the *rho_spectral* feature had the largest average contribution (65%) to the prediction score as opposed to the *rho_chaos* score which had the lowest impact (5%) (Fig. 6A). This is also consistent when stratified by context in animal and plant-based datasets (Fig. 6B and Supplementary Fig. 25). Interestingly, FTICR-based datasets have a higher variability in the spectral feature importance than Orbitrap-based datasets in animal-based datasets.

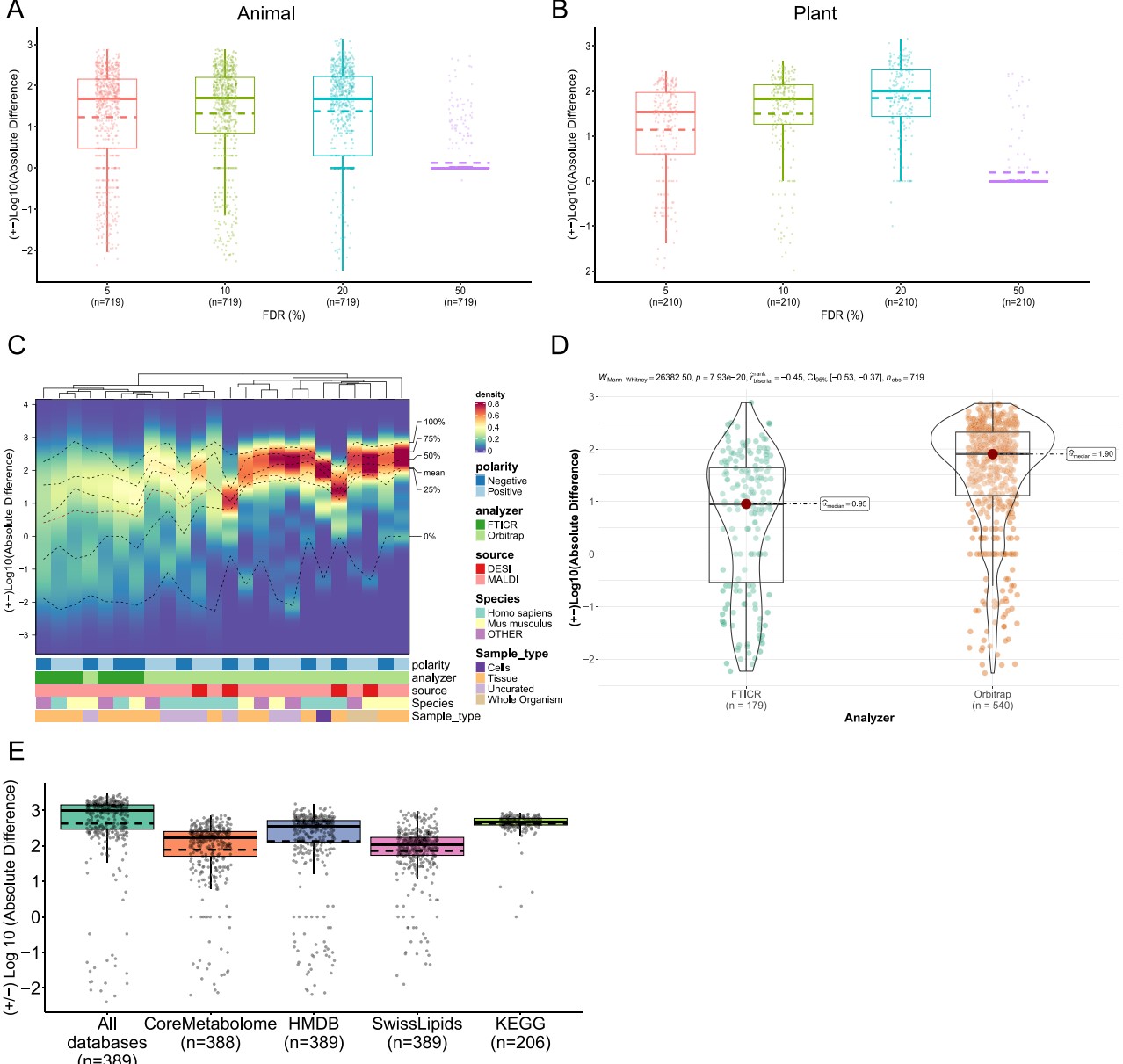

**Fig. 4 | Assessment of annotation coverage for METASPACE-ML vs rule-based approach. A, B** Boxplots showing the difference in the number of annotations captured by METASPACE-ML relative to MSM at different FDR thresholds for animal (**A**) and plant (**B**) testing datasets. Each dot represents a dataset, and the y-axis shows the Log10 absolute difference, where negative values indicate MSM had more annotations than METASPACE-ML and vice versa. Solid and dashed lines represent the median and mean, respectively. The number of datasets is displayed below the *x*-axis labels. **C** Density heatmap showing the distribution of Log10 absolute difference scores across datasets for each context in animal testing datasets. Each column represents a context described by its metadata, with bars colored by different classes. The *y*-axis shows Log10 absolute difference scores, and the color gradient represents density. Columns are hierarchically clustered using the Kolmogorov-Smirnov statistic. Dotted lines indicate different quantiles and the mean. **D** Box-violin plot showing Log10 absolute difference score distributions for

FTICR and Orbitrap animal testing datasets. The *p*-value and test statistic of a two-tailed Wilcoxon rank-sum test are shown above the plot, with the number of datasets per analyzer in parentheses below the x-axis labels. **E** Boxplot showing the distribution of the difference in number of annotations captured by METASPACE-ML relative to the MSM method across different databases at FDR 10%. A dot represents a dataset from 206 animal testing datasets (see methods). The *Y*-axis represents the Log10 absolute difference, where negative values are those where MSM had more annotations than METASPACE-ML and vice-versa. The median and mean are represented by solid and dashed lines, respectively. In (**A**, **B**, and **D**), boxplots' bottom and top edges represent the 25th and 75th percentiles, with the median (50th percentile) line inside the box. Whiskers extend to the minimum and maximum values within 1.5 times the interquartile range from the quartiles; the minimum and maximum values are represented by the extent of the jittered data points.

To further see the effect of different features and evaluate the discriminatory power of METASPACE-ML vs the rule-based approach, we visualized the target and decoy ions for each dataset by performing UMAP (Uniform Manifold Approximation and Projection) of the ions using all 5 features and visually assessed how different scoring approaches separate targets from decoys. Examining the UMAP for a

mouse brain dataset (https://metaspace2020.eu/dataset/2016-09-22_11h16m09s) which we later used for bulk validation, we observed that the METASPACE-ML score better reflects the target-decoy separation than MSM (Fig. 6C). Taking a closer look at the pairwise relationships between the 5 features, we observed that the main differences are driven predominantly by *rho_spatial* and *rho_spectral* scores, where

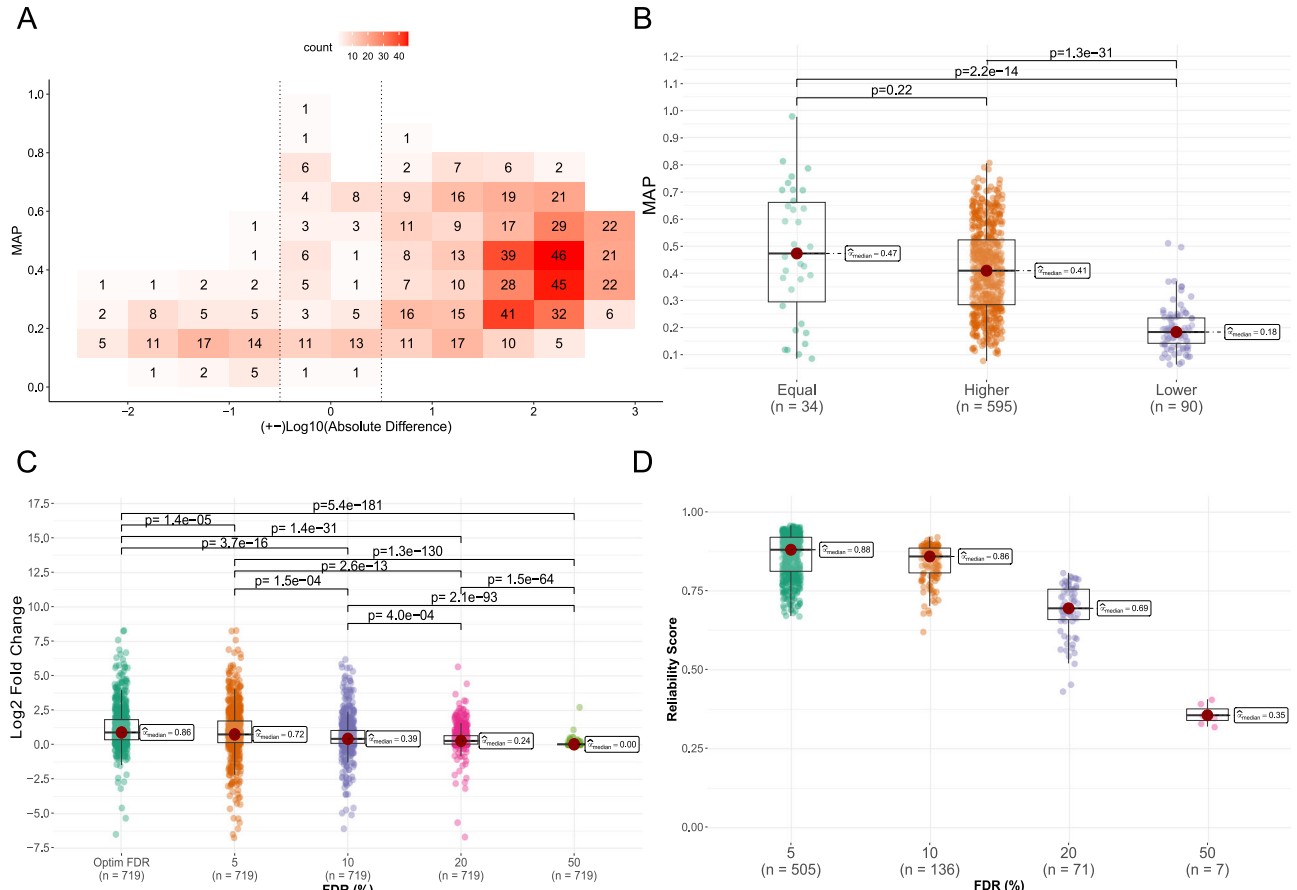

**Fig. 5 | Evaluation of reliability of annotations predicted by METASPACE-ML.**
**A** 2D binned plot showing the relationship between the increased number of annotations compared to MSM (Log10 absolute difference) and MAP scores. Each bin, with a width of 0.5 and length of 0.1, displays the count of animal testing datasets, with a color gradient indicating density. Bins within dotted lines represent datasets with ≤0 Log10 absolute difference scores. **B** Boxplot of MAP scores for animal testing datasets, grouped by whether METASPACE-ML had equal, higher, or lower annotations compared to MSM. **C** Boxplot of Log2 folds changes in the number of annotations of METASPACE-ML relative to MSM across all animal testing datasets, for different FDR thresholds. Exact *p*-values from a two-tailed Wilcoxon rank-sum test in (**B**) and (**C**) are shown above each comparison. **D** Boxplot of reliability scores across all animal testing datasets, for optimal FDR thresholds. In (**B**–**D**), boxplots' bottom and top edges represent the 25th and 75th percentiles, with the median (50th percentile) line inside the box. Whiskers extend to the minimum and maximum values within 1.5 times the interquartile range from the quartiles; the minimum and maximum values are represented by the extent of the jittered data points.

targets don't have low spectral scores, and the majority of decoys tend to have low spatial scores (Supplementary Fig. 26) which is concordant with the feature importance scores.

Based on the previous results, it may look like METASPACE-ML provides more true positives at the expense of increased false positives, but as we showed earlier using the Fbeta measure in the reliability scores, the optimal balance between false positives and false negatives is still preserved in favor of METASPACE-ML. This is also illustrated by the AUC-ROC and area under the precision-recall curve of the brain dataset compared to the rule-based approach and each individual feature (Fig. 6D and Supplementary Fig. 27).

Finally, we zoomed into the type of decoy adducts, and for each adduct type, we checked whether it had more decoy ions at FDR < 10% based on METASPACE-ML compared to the rule-based approach across animal-based testing datasets. Those decoy adducts represent the METASPACE-ML false positives. We did the same using MSM and then compared the distribution of those adducts' exact mass between METASPACE-ML false positives and MSM false positives, and we found that METASPACE-ML false positives had significantly (*p*-value < 0.05) lower mass compared to the false-positives picked up by rule-based approach (Supplementary Fig. 28).

## METASPACE-ML captures ions of low intensity and from biologically-relevant classes

Having established the improved performance of METASPACE-ML in delivering more annotations and providing higher ranking quality, we examined the properties of target ions that are exclusively annotated by METASPACE-ML compared to the rule-based approach. Using the list of animal datasets, we observed that in nearly all the datasets with annotations at FDR 10% (698/720), compared to MSM, newly-found annotations by METASPACE-ML had significantly (*p*-value < 0.001) lower intensities (Fig. 7A and Supplementary Fig. 29 for a detailed example). This represents a substantial advantage and may be especially useful in annotating biologically relevant ions corresponding to low-concentration metabolites, and thus turning acquired data into actionable hypotheses. The pattern is also consistent if we stratify the comparison per context as well (Fig. 7B and Supplementary Fig. 30), where DESI-based datasets have the lowest difference in intensities between METASPACE-ML and MSM (Fig. 7B).

Next, we investigated which metabolic classes are represented among ions annotated by METASPACE-ML only, at FDR 10%. By using the one-tailed Fisher exact test and the fold enrichment as a proxy for an enrichment score (see "Methods"), we found that fatty acyls,

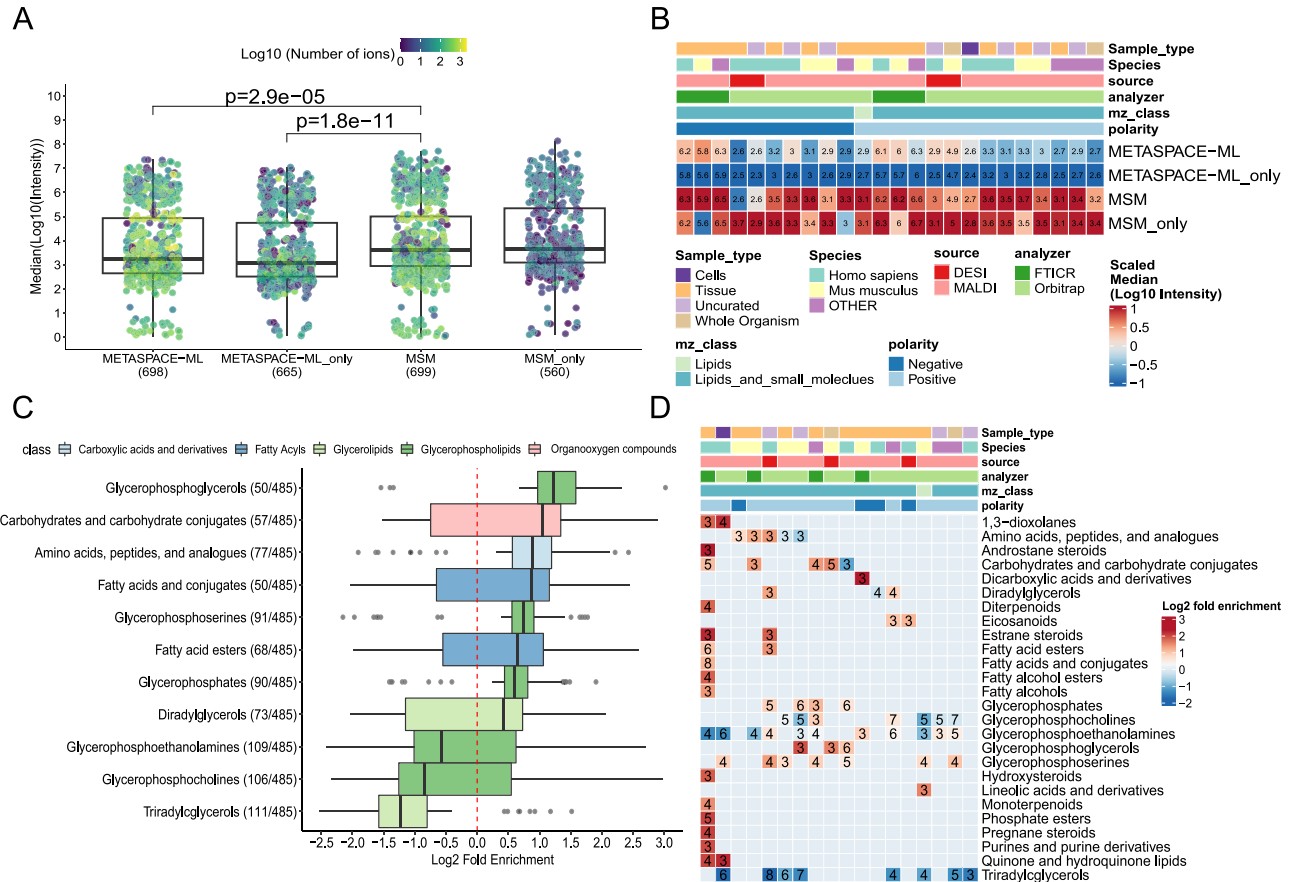

**Fig. 6 | Assessment of the target-decoy separation and feature importance.**
**A** Ridge plot showing the density of SHAP impact contribution scores (see "Methods") for each of the five ion features used for METASPACE-ML model training across all testing datasets. Features are displayed on the y-axis, and the SHAP contribution scores are displayed on the x-axis. Quartile lines are displayed for each ridgeline, and colors represent the area under each ridgeline for each of the 4 quartiles. **B** Density heatmaps show the distribution of SHAP impact contributions scores across datasets for each context in animal-based testing datasets and faceted by each of the five features. Each column represents a context which is described by its constituent metadata as bars colored by the classes in each

metadata variable. The y-axis shows the SHAP impact contributions scores, and the color gradient represents their density. Columns are hierarchically clustered using a distance metric based on the Kolmogorov-Smirnov statistic. **C** UMAP for both target and decoy ions for one of the brain datasets used for the LC-MS bulk validation (https://metaspace2020.eu/dataset/2016-09-22_11h16m09s). A dot represents an ion and is colored by whether it's a target or decoy (top left), the MSM score (top right), and the METASPACE-ML score (bottom left). **D** ROC curves for the same dataset as in (**C**) showing sensitivity and False Positive Rate (FPR) using the METASPACE-ML and MSM scores as well as each of the five constituent ion features. Curves are colored by the scores they correspond to.

carbohydrates, amino acids, and most glycerophospholipid subclasses were better annotated by METASPACE-ML in animal datasets (Fig. 7C), while lactones, keto acids, and carboxylic acids were better annotated by METASPACE-ML in plant datasets (Supplementary Fig. 31). If we stratify the enrichment results by context, we find classes such as glycerophosphocholines, glycerophosphoethanolamines and amino acids showing context-specific overrepresentation compared to other contexts, especially in animal datasets. On the other hand, triacylglycerols show significant underrepresentation in all contexts in both animal and plant-based datasets (Fig. 7D and Supplementary Fig. 32), indicating that the rule-based approach has a high recall for such classes which explains why they are not picked up exclusively by METASPACE-ML.

## LC-MS/MS bulk validation of newly annotated ions

In order to add another layer of validation of the ions annotated exclusively by METASPACE-ML, we considered the MALDI-imaging data together with the bulk LC-MS/MS validation data used in the original MSM publication and performed a comprehensive identification of all ions in the LC-MS/MS data[6]. In total, we considered 10 mouse brain MALDI-FTICR imaging datasets (Supplementary Data 8). First, we compared the ranking quality and annotation coverage for each of the

datasets, and we observed that, on average, METASPACE-ML achieved higher MAP scores than the rule-based predecessor (0.57 vs 0.43) in 9/10 of the datasets (Fig. 8A) and METASPACE-ML captured a median of ~ 63 and 30 more annotations than the rule-based approach at 5% and 10% FDR, respectively (Fig. 8B). Then, we compared the ions identified from the matched bulk LC-MS/MS data (Supplementary Data 9) with those annotated by either the rule-based approach or METASPACE-ML in the imaging MS data by matching sum formulas and adducts that are MS/MS validated to the annotations that have FDR < 10% for each approach. Illustrating it for one dataset that was in the same project as the one used in Fig. 6C (https://metaspace2020.eu/dataset/2016-09-22_11h16m17s), METASPACE-ML shows a true positive rate (TPR) of 91% overall (Fig. 8C) and if we only consider ions exclusively captured by either of the approaches, we observe a 72% vs 11% TPR captured for METASPACE-ML and the rule-based approach, respectively (Fig. 8D, E). Similar trends were obtained for the other 9 datasets with the diagnostic metrics, including TPR, FPR, and FNR shown in Supplementary Fig. 33.

Altogether, this confirms that METASPACE-ML achieves a high true positive rate in predicting ions identified in the corresponding bulk LC-MS/MS data and that METASPACE-ML outperforms the rule-based predecessor.

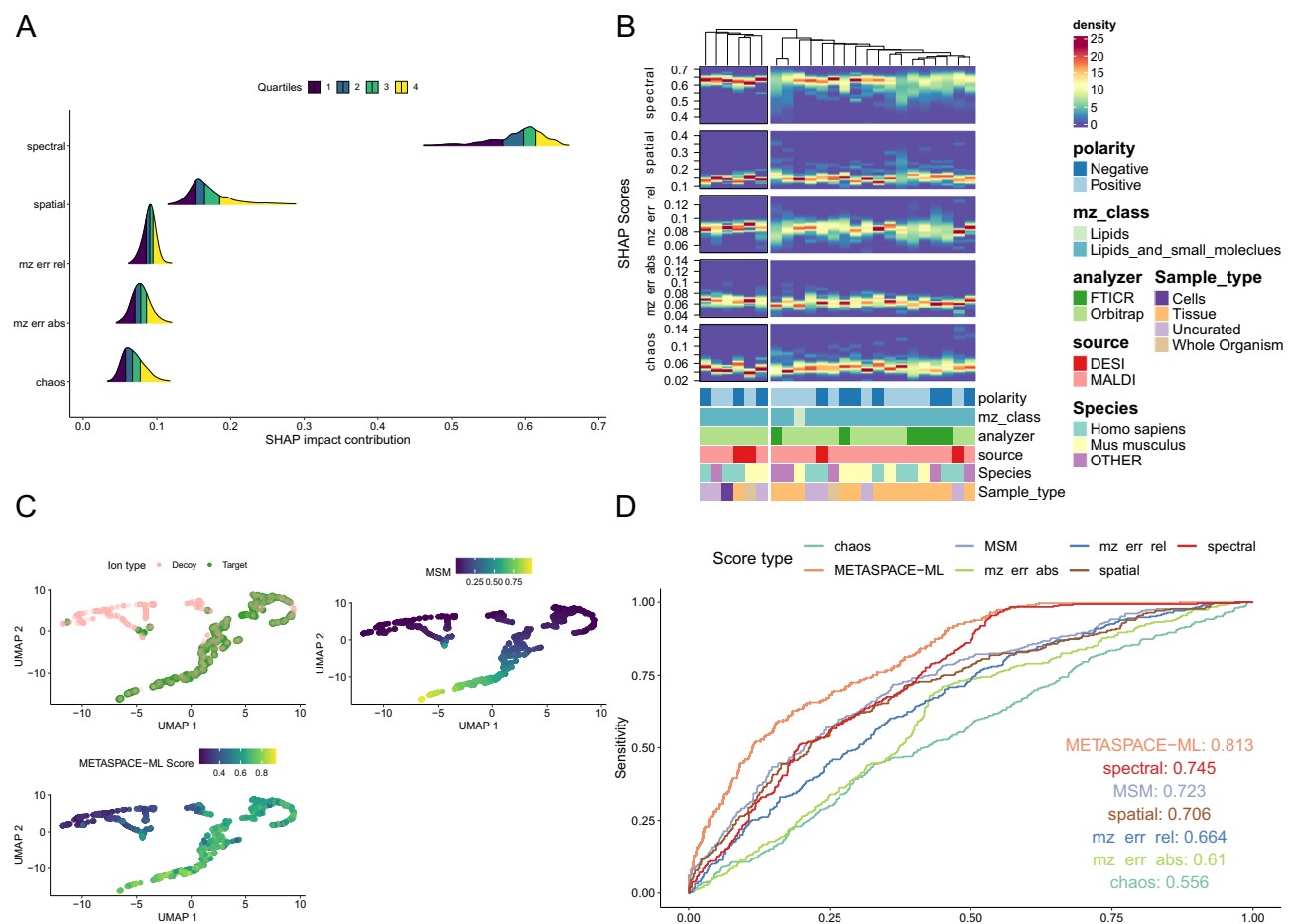

**Fig. 7 | METASPACE-ML captures low-intensity ions and biologically-relevant metabolites. A** Boxplots showing the median intensity distribution for each animal testing dataset across total and unique annotations by METASPACE-ML or MSM at FDR 10%. Each dot represents a dataset, colored by the Log10 number of annotations. *X*-axis labels indicate annotation approaches, with "only" denoting exclusive annotations. *The Y*-axis shows the median Log10 intensity for all ions per dataset. Significant *p*-values ($p < 0.05$) from a two-tailed Wilcoxon rank-sum test are shown above comparisons. **B** Heatmap displaying median Log10 intensity per context across animal testing datasets. Columns represent contexts described by metadata, with colors indicating classes. Rows show approaches from (**A**). Color gradients correspond to the median Log10 intensity. **C** Overrepresentation analysis using one-tailed Fisher's exact test for datasets to identify enriched metabolite/lipid classes in ions exclusively captured by METASPACE-ML at FDR 10%. Log2 fold

enrichment is on the *x*-axis, and HMDB metabolite subclasses are on the *y*-axis. Boxplots are colored by parent class, with the number of datasets in parentheses. Only terms with significant enrichment ($p < 0.05$) in at least 10% of datasets are shown. **D** Heatmap showing overrepresentation analysis results per context in animal testing datasets. Columns represent contexts described by metadata, with colors indicating classes, and rows show significantly enriched metabolite classes. Color gradient corresponds to Log2 fold enrichment, with labels indicating the number of datasets per context. In (**A**) and (**C**), boxplots' bottom and top edges represent the 25th and 75th percentiles, with the median (50th percentile) line inside the box. Whiskers extend to the minimum and maximum values within 1.5 times the interquartile range from the quartiles; the minimum and maximum values are represented by the extent of the jittered data points.

## Discussion

The demonstrated performance of METASPACE-ML relies on three methodological advances compared to the approach we proposed earlier[8]. First, we used the Gradient Boosting Decision Trees (GBDT) approach, formulated as a ranking model. This proved particularly valuable as GBDT is known to be robust to noisy outliers, which can significantly impact the separation of target and decoy ions. Moreover, we proposed two new features estimating absolute and relative m/z error for centroided data, which helped improve the accuracy of the model predictions, albeit slightly. Given the increasing use of centroided data in the field allowing efficient compression, these scores hold potential for wider application beyond METASPACE-ML. We also introduced an expert-curated metabolome database, CoreMetabolome, which resulted in higher MAP scores compared to more general databases.

By utilizing the largest collection of public imaging MS data, we were able to select large numbers of datasets for training and

evaluation. Importantly, we defined the relevant contexts covering combinations of technology platforms and protocols with types of samples. This helped select datasets representing various contexts with a fixed context size to allow comparative evaluation. Moreover, we trained two independent models for plant and animal datasets, each with a unique combination of contexts encompassing both analytical and biological metadata to provide a more granular approach for end-users to align and cross-reference their datasets with the training and evaluation datasets. Accordingly, to help users align their datasets to the context-based results discussed above, we have developed a separate web app (https://t.ly/q-nb5) so that users can get more information about which datasets the model was trained and tested on, as well as being able to evaluate the performance of the model on those datasets. In addition, we introduced quality control criteria, in particular, to exclude non-centroided datasets submitted to METASPACE by mistake (see "Methods"). Non-centroided datasets cannot be properly annotated as implausible decoy ions have the

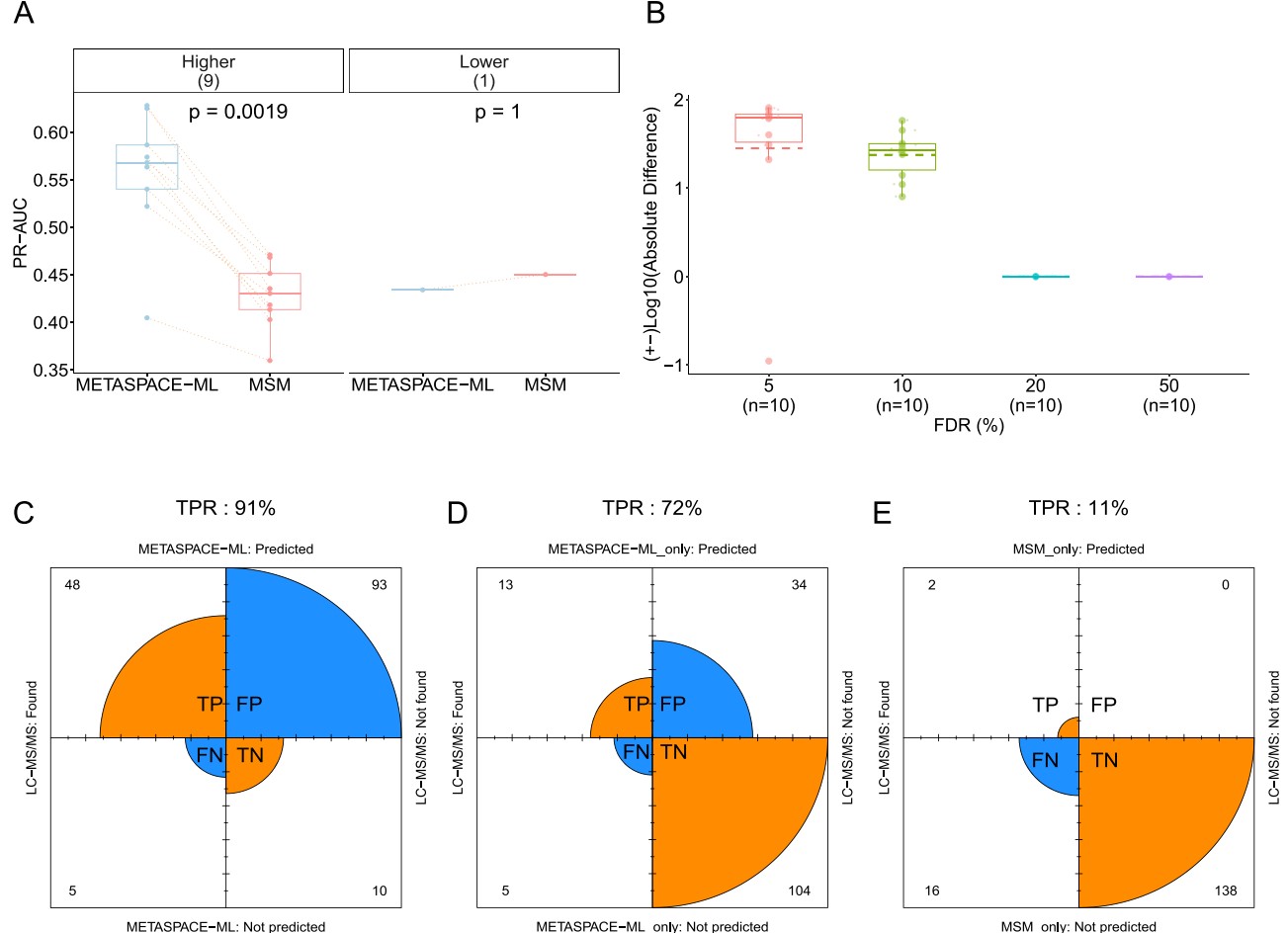

**Fig. 8 | METASPACE-ML achieves a high true positive rate as compared to ions identified with bulk LC-MS/MS. A** Paired boxplot showing PR-AUC per dataset for 10 brain datasets (source data are provided in Supplementary Data 8) from bulk LC-MS/MS re-analysis. The x-axis represents the method, and the y-axis the PR-AUC scores. Each dot represents a dataset, with orange dotted lines connecting the same datasets. Boxplots are grouped by AUC score differences (Delta), indicating higher AUC in METASPACE-ML relative to MSM. Dataset counts per group are shown in parentheses. Exact p-values are from a two-tailed paired Wilcoxon signed-rank test. **B** Boxplots showing Log10 numbers of annotations by METASPACE-ML relative to MSM across different FDR thresholds for datasets in (**A**). Each dot represents a

dataset; the y-axis shows the Log10 absolute difference, with negative values indicating MSM had more annotations. Solid and dashed lines represent the median and mean, respectively. Dataset counts are below the x-axis labels. **C**, **D**, and **E** Fourfold plots for ions captured by METASPACE-ML (**C**), exclusively by METASPACE-ML (**D**), and exclusively by MSM (**E**), compared to LC-MS/MS verified ions for one brain dataset (https://metaspace2020.eu/dataset/2016-09-22_11h16m17s). True and false hits are orange and blue, respectively, with arc size corresponding to ion counts shown in each quadrant. True Positive Rate (TPR) is displayed above each plot.

tendency of m/z-overlapping with the peak shoulders, thus leading to low-to-no annotations at the low FDR levels. Newly submitted datasets that do not adhere to the defined quality criteria can be flagged, and the corresponding users can be notified prior to processing.

A key aspect of METASPACE-ML and its rule-based predecessor is the control of the False Discovery Rate (the ratio of false positives) in the provided results. This helps an end-user assess the reliability of annotations and select the level of confidence in the results by choosing the FDR threshold. Here, with METASPACE-ML, we improved the FDR estimation procedure (see "Methods") and proposed a reliability score that helps the end user identify the optimal FDR threshold that optimizes both precision and recall. Similarly, we provide a more detailed diagnostic of the contribution of each feature to the final prediction score for each annotation which adds an additional layer of explainability of the annotation results.

In addition to achieving high reliability of predictions, increased annotation coverage, and better ranking quality, we show that the ions exclusively annotated by METASPACE-ML tend to have lower intensities compared to the rule-based predecessor, which is especially useful in annotating biologically relevant ions corresponding to low-

concentration metabolites. This is further corroborated by the significant enrichment of essential structural metabolic classes that are pivotal in various metabolic pathways. Furthermore, we showed that many ions exclusively captured by METASPACE-ML in mouse brain datasets match with their corresponding LC-MS/MS bulk validation, demonstrating the improved sensitivity of METASPACE-ML compared to its rule-based predecessor.

Despite the notable performance and annotation coverage achieved by METASPACE-ML compared to the rule-based MSM approach, we acknowledge its limitations. Most of the public datasets used for training and testing were acquired with either FTICR or Orbitrap mass analyzers. With the recent surge in the popularity of Quadrupole Time of Flight (QTOF)-based systems, an increasing number of QTOF-based datasets are being acquired. However, these datasets are currently underrepresented in the proposed model due to the lack of publicly available data needed for statistically reliable model evaluation. We anticipate that in the future, our models can be specifically trained for QTOF analyzers as more datasets become publicly available. In terms of used features, *rho_chaos*[6] had only a minor impact on the model performance (Fig. 6A), and target-decoy

separation was mainly driven by *rho_spectral*, which demands re-defining the quantification of spatial informativeness and further exploration for potential new features for training. Examining Feature scores, it was observed that in some datasets, the MSM approach can capture more annotations compared to METASPACE-ML. We observed that some annotations were captured by MSM but not by METASPACE-ML due to MSM's sensitivity to its feature scores, particularly the *rho_spatial* score. In certain datasets, the target ion may have a high *rho_spatial* score, while decoys have low *rho_spatial* scores, resulting in a high MSM rank and low FDR for the target ion. Conversely, METASPACE-ML prioritizes *rho_spectral* scores over *rho_spatial* scores, which can cause higher FDR for these targets as the decoy scores are closer to the target ion scores. Regarding decoys, as discussed earlier[6], the way of producing decoy ions is key, and new data shows their heterogeneity in terms of similarity to the target ions (Fig. 6C). Further investigations are needed into how this affects FDR estimation and in finding the most reliable way of decoy generation. Lastly, although the training datasets were selected to be representative of a large number of public datasets from 47 labs, this selection is biased towards public datasets represented in METASPACE. This warrants further work for evaluating METASPACE-ML for less common data, e.g., from cultured cells in single-cell metabolomics, or data from industrial labs, where the ability to deposit data publically may be limited. Finally, while the introduced CoreMetabolome database does offer improved performance, more rigorous and automated methods for generating curated databases are desired.

Due to its flexible architecture, METASPACE-ML can include additional features. Of particular interest is the integration of Collisional Cross-Section (CCS) values, enabling automated use of ion mobility separation able to resolve molecular isomers and isobars[11]. Furthermore, METASPACE-ML can incorporate the Kendrick mass defect or features quantifying other phenomena e.g., co-detection of characteristic in-source fragments or biochemically related molecules. Since the model can be trained or fine-tuned on any imaging MS data, one can envision the development of context-specific, technology-specific, or lab-specific models as compared to generalized models.

METASPACE-ML provides a general and flexible framework for evaluating how various signal-processing steps affect metabolite annotation. One can envision using this approach to maximize the extraction of molecular information by optimizing spectral recalibration, spectral alignment, and various aspects of transient processing for Fourier-transform ion cyclotron resonance (FTICR) or Orbitrap, as well as denoising Quadrupole Time-of-Flight (QToF), peak picking, centroiding, and the denoising of ion images.

With the growing number of public imaging MS datasets available in METASPACE, we can train future generations of METASPACE-ML by including more datasets and covering more contexts. Importantly, we envision training custom METASPACE-ML models for specific contexts or data providers, thus achieving the best metabolite annotation for their data. Finally, the increased computational efficiency of METASPACE-ML, together with the use of the flexible, serverless computing framework Lithops, will allow us to reprocess public historic METASPACE datasets (over 10,000 as of January 2024) to increase the value of this public collection and shed light on the unannotated 'molecular dark matter' in this big data.

## Methods

### Metadata curation of public datasets
In order to better classify and contextualize the datasets for selection, existing metadata fields and additional classifications were introduced. First, the metadata of a total of 9251 public datasets was downloaded using the METASPACE API (https://metaspace2020.readthedocs.io) on September 13, 2023. The metadata associated with each dataset includes sample information, sample preparation, MS analysis settings, annotation configuration, publication status, and project association. Using the sample information for each dataset, the organism field was manually standardized to either a genus/species level classification (e.g., Human to *Homo sapiens*, Mouse to *Mus Musculus*) whenever possible. In total, 275 different organism names were filtered down to 143 genus/species (Supplementary Data 1). Moreover, a kingdom metadata field was added based on the previous organism classification to further classify the organism into one (Animal, Plant, Bacteria, Fungi, Protista, Undefined) whenever possible (Supplementary Data 1). For datasets that are part of a project, using information from sample information, sample preparation, and manual inspection, an additional metadata field (Sample type) was added to classify datasets into one (Tissue, Cells, Whole Organism, Spheroids, Environmental, Spots). Keywords like biopsy or tissue section were used to classify datasets as tissue, while keywords like single cells, cell-monolayer, co-culture, and cultured cells were used to classify datasets into cells (Supplementary Data 2). Most sample type classification was curated for datasets associated with a project, however, since most datasets submitted to METASPACE are tissues, we considered the acquisition geometry of the ion images as a filter to classify non-project associated datasets into tissue based on the assumption that images with irregular geometries are mostly tissues. Ion images where the total pixel count is not equal to the product of the number of pixels in x and y coordinates are classified as irregular (Supplementary Data 3). Datasets that are not project-associated and don't have an irregular geometry were classified as uncurated. Finally, datasets were also classified based on their m/z range, where datasets having maximum m/z ≤ 400 are classified as small molecules, and those having minimum m/z > 500 are classified as Lipids. The rest of the datasets with min m/z < 500 and max m/z > 400 are classified as "Lipids and small molecules" (Supplementary Data 2).

### Exclusion of non-centroided datasets
METASPACE requires users to submit centroided datasets. However, some users, by mistake submit non-centroided profile datasets, which can lead to overselection of peaks. Such cases lead to suboptimal annotation because of the increased likelihood of isotopes of decoy ions matching shoulders of peaks or noise peaks. So, we have developed a strategy to identify and exclude such datasets.

To avoid having to check the spectra for each pixel in each dataset among thousands of public datasets, for each dataset, we considered the top $n = 50$ pixels with the highest number of non-zero intensity m/z peaks. For each of those 50 pixels, we consider a vector of non-zero m/z values denoted by $\mathbf{m} = [m_1, m_2, m_3, ..., m_N]$ where $m_i$ represents i'th non-zero m/z value with $N$ peaks altogether in the spectrum. For the m/z tolerance *m/z_tol_ppm* used in METASPACE-ML defined in ppm (here we use 3 ppm), we can calculate the proportion of consecutive m/z peaks that are found within a given ppm by comparing the differences in m/z between consecutive peaks according to Eq. (1) below:

$$proportion\ overlap = \frac{1}{N-1} \sum_{i=1}^{N-1} I\left((m_{i+1} - m_i) \geq \frac{m/z\_tol\_ppm}{10^6}\right)$$

(1)

where $I(.)$ is the indicator function returning 1 if the argument is true and 0 otherwise, and $N$ is the number of non-zero m/z peaks. Then, for each dataset, we take the average proportion to overlap over the aforementioned 50 pixels as a heuristic score to flag datasets for exclusion. Based on the distribution of those scores overall public datasets, we considered datasets with a score > 0.5 for exclusion.

### Exclusion of low-quality datasets
We have developed the following criteria to exclude low-quality datasets from training and testing sets: (1) The number of annotations at FDR 20% is less than 10 for each possible target adduct and annotation database combination, which resulted in the exclusion of 1418

datasets; (2) The median (across pixels) number of m/z peaks with non-zero intensity > 50,000, this led to the exclusion of 6 datasets; (3) The *proportion_overlap* score > 0.5 (see "Methods") which led to the exclusion of 127 datasets. More information about the quality filters for all considered datasets can be found in Supplementary Data 4.

### Context-dependent selection of training and testing datasets

To ensure a representative selection of different datasets from the Metaspace knowledge base, we first categorized all public datasets into contexts based on existing and newly added metadata (see metadata curation). The datasets were classified based on possible combinations of the following 6 variables: polarity, ionization source, mass analyzer, m/z class, sample type, and species. Ideally, a given context is defined by all 6 variables, however in very few cases some datasets might match to multiple contexts in cases where sample type is not well defined. In such cases, the matching will be made so that it considers combinations of the fixed 5 variables with all possible sample types (See Context explorer Shiny app for more information). Moreover, the selection of training and testing datasets were performed separately for animal and plant-based datasets, which constitute 92% of all public datasets. For each kingdom, the species variable was first grouped so that species with cumulative frequency of the bottom 10% will be grouped as "OTHER". Also, we only considered the following sample types (Tissue, Whole organism, Cells, and Uncurated). To ensure the statistical reliability of the predictions, we only selected contexts with at least 45 datasets, where 30 would be used for testing and the rest for training and cross-validation. Given that the number of training datasets might be scarce for the model and to test the effect of underfitting vs overfitting, 5 different models were trained of varying numbers of datasets per context, starting from a minimum of 10, up to a maximum of 50 datasets per context. Accordingly, for some models, certain contexts are not represented due to the insufficiency of available datasets to cover both testing and training.

Once the number of datasets for each context was determined, the next step was to optimize the sampling procedure to maximize the diversity of datasets within each context. Using information about the institute / lab (group) that submitted the dataset, the project ID (for project-associated datasets), and the submission day, we tailored the sampling procedure based on the following assumptions: (1) datasets submitted by the same group and project are more homogenous (2) datasets submitted in the same day are more likely to be biological / technical replicates and (3) datasets from different groups are more heterogeneous. For datasets not associated with a project, a pseudo-label for the project was created based on the submitter name and day of submission. Given the number of required datasets per context, a recursive sampling procedure is performed to iteratively sample datasets so that at least one dataset per project-group combination is selected from the context-specific datasets, in cases where the size of the sampling pool is more than the required datasets, the datasets are weighted and ranked by both the relative size of project-group combination as well as their Shannon's entropy. Then, the remaining datasets are selected from the top-ranked list so that project-group combinations with maximum entropy and highest relative size are used to randomly select the remaining datasets. Based on this previous sampling procedure, 30 testing datasets per context were selected first for each kingdom, and then those datasets were removed from the selection pool for the training datasets to make sure that training and testing datasets were mutually exclusive. The selection of 30 datasets per context was chosen to strike a balance where including more datasets could lead to less diversity of contexts, whereas including fewer datasets might risk compromising statistical reliability due to a smaller sample size potentially leading to less robust evaluation. The final list of selected datasets for each context and kingdom can be found in Supplementary Data 5.

### Processing training data using the rule-based METASPACE and CoreMetabolome

The selected training datasets were then reprocessed on the META-SPACE server using the rule-based approach as previously described[6] with minor modifications to their configuration file (Supplementary Note S1). The datasets were annotated against the CoreMetabolome database. The CoreMetabolome database is a molecular database specially designed and developed by us for METASPACE annotation. CoreMetabolome was designed to be large enough for untargeted spatial metabolomics, include only chemically plausible molecules, prioritize endogenous molecules over exogenous molecules, and include the molecules from primary metabolic pathways that are commonly abundant in different types of samples. HMDB (version 4) and KEGG were used as input databases and curated manually by an experienced chemist and mass spectrometrist. Supplementary Note S2 contains detailed information on the curation process.

### New ion scores quantifying the m/z error from the centroided data

In addition to the scores used in the Metabolite Signal Match score (MSM) (spatial isotope *rho_spatial*, spectral isotope *rho_spectral*, and spatial chaos *rho_chaos*), we have introduced two new scores: *m/z error abs* (absolute m/z error) and *m/z error rel* (relative m/z error). These scores quantify the error in estimating the m/z value for an ion of interest compared to its theoretically defined value as follows:

$$\underline{m} = \frac{\sum_{p=1}^{n} m_p I_p}{\sum_{p=1}^{n} I_p}, \quad (2)$$

$$m/z\_error\_abs = 1 - |\underline{m}_{i=1} - \hat{m}_{i=1}| \quad (3)$$

$$m/z\_error\_rel = 1 - \left| \frac{\sum_{i=2}^{T} \left( (\underline{m}_i - \hat{m}_i) - (\underline{m}_{i=1} - \hat{m}_{i=1}) \right) * \hat{I}_i}{\sum_{i=2}^{T} \hat{I}_i} \right| \quad (4)$$

where, for a given ion, $m_p$ is the m/z value in pixel $p$ of the respective ion image, $I_p$ is the corresponding intensity in pixel $p$, and $n$ is the total number of pixels in a given ion image. Moreover, $\hat{m}_i$, $\underline{m}$ and $\hat{I}_i$ are the theoretical m/z value, the observed mean (across pixels), and the theoretical relative intensity of the $i$'th isotopic peak of that ion, respectively.), $T$ is the total number of isotopic ion peaks considered (in METASPACE-ML we use $T = 4$).

While *m/z error abs* quantifies the m/z error between observed and theoretical only in the first isotope, the *m/z error rel* quantifies the difference between observed and theoretical values for 2'th to $T$'th isotopes relative to the difference between observed and theoretical in the first isotope.

### FDR estimation

In the rule-based approach[6], the target-decoy strategy was formulated so that for an MSM threshold, the ratio of positive decoys to all positives provides an estimate for FDR. To reduce the variability introduced by the random choice of decoy adducts, the decoy adducts were sampled $S_D = 20$ times for each target adduct, taking the median value across $S_D$ rankings for each formula before applying monotonicity adjustments. In METASPACE-ML, we also sample decoy adducts $S_D$ times, yet we propose to use a single weighted ranking where decoys are weighted with $1/S_D$. First, for a database and each target adduct, both target ions and all sampled decoy ions are sorted in descending order based on the model prediction score. For each rank threshold $i$ of sorted ions, we calculate $T_i$ and $D_i$ which are the numbers of targets and decoys, respectively, with ranks smaller or equal than $i$.

The FDR value for an ion with the rank threshold $i$ was defined as

$$FDR_i = \frac{(D_i+1)/S_D}{(T_i+1)+((D_i+1)/S_D)} \qquad (5)$$

A pseudocount of 1 was added to both $T_i$ and $D_i$ as per the rule of succession to avoid misleading 0% FDR and for a better estimate of the mean of targets and decoys which have a binomial-like distribution.

In summary, we introduced the following changes to the FDR estimation compared to the rule-based approach[6]: (1) Using a single weighted ranking, where decoys are given $1/S_D$ of the weight, (2) Using a single selection of $S_D$ random decoys per formula which are shared between all FDR rankings, (3) Allowing for calculation of continuous FDR values for each ion instead of snapping FDRs to fixed thresholds (5%, 10%, 20%, 50%), and (4) Introducing a rule of succession where a pseudocount of 1 is added to the number of targets and decoys.

These changes increase the computational performance. In the rule-based approach, $S_D$ random decoys would be sampled from a set of implausible adducts for each formula and target adduct (e.g., + H, + Na, + K). In METASPACE-ML, we propose instead to randomly sample $S_D$ decoy adducts per target formula and share them across all possible target adducts for that formula. This change should not affect the FDR rankings as they are statistically independent. However, this allows to reduce the calculations of scores as it produces fewer decoy ions overall. To show an example of calculated FDR alongside the scores, we provide (Supplementary Data 10) which presents the scores for both MSM and METASPACE-ML for both targets and decoys, along with their corresponding FDRs. For a given FDR threshold, such as 10%, a true positive is defined as a target ion with an FDR < 10%, while a false positive is a decoy ion with an FDR < 10%. This table includes the input data used to estimate the FDR for a specific group (+ Na adducts) within a particular dataset (https://metaspace2020.eu/dataset/2018-12-14_16h34m31s).

## Reliability score for target-decoy annotations

In order to assess the reliability of the results provided in the target-decoy-based annotation and help select the optimal FDR threshold (e.g., from 5%, 10%, 20%, and 50%) we formulated a reliability score *reliability_score*. This score can help minimize false positives and maximize annotation coverage. Accordingly, we considered the concept of the F-score that estimates the balance between precision and recall. The most popular F-score is the F1-score which is the harmonic mean of precision and recall. However, given the inherent imbalance between target and decoy ions, as we sample $n = 20$ decoy ions for each target ion for robustness, there are more negative than positive instances in our data to be ranked, and thus precision is more important than recall given the higher number of false positives. Thus, we chose to optimize the F-beta score, which weights the contribution of precision vs recall based on the value of beta. Here, we choose beta = 0.5 to put more weight on the precision while still considering the recall; see Eq. (6). Then, using the Cutpointr R package[12], we calculated the F-beta score for the annotation results where target and decoy ions are considered as positive and negative instances, respectively. For each dataset, we consider a vector for all possible FDR cutoffs denoted by $\mathbf{f} = [f_1, f_2, f_3, ..., f_K]$ where $f_i$ represents the i'th FDR cut-off sorted ascendingly for $K$ possible cutoffs. For each FDR cut-off, we get a corresponding F-beta score denoted as $\mathbf{b} = [b_1, b_2, b_3, ..., b_K]$ where $b_i$ represents i'th F-beta score for K possible FDR cutoffs. Then, for each of the 4 default FDR thresholds considered in METASPACE (5%, 10%, 20%, and 50%), we calculate the reliability score in Eq. (8) as the product of two terms: (1) a ratio between the F-beta score at the FDR-cutoff closest to the chosen fixed threshold relative the maximum F-beta score and (2) complement of the optimal FDR-cutoff (FDR-cutoff at maximum F-beta score). The first term measures how far is the F-beta score at the chosen FDR threshold from the maximum

possible F-beta score and has a scale of [0,1]. The second term penalizes the overall score based on the optimal FDR value: the lower the optimal FDR, the lower the penalty. Finally, since we calculate the reliability score for all four METASPACE-default FDR thresholds, we can select the minimum FDR threshold that has the maximum reliability score as the optimal FDR threshold at which the annotations are most reliable.

$$F\,beta = \frac{(1+\beta^2)*precision*recall}{\beta^2*precision+recall}; \beta = 0.5 \qquad (6)$$

$$optim_{fdr} = f\,[argmax_i\,b[i]] \qquad (7)$$

$$reliability\_score = \frac{b[argmin_i|f[i]-FDR_{METASPACE\_thresh}|]}{max(b)} \\ *(1-optim_{fdr}) \qquad (8)$$

where, $\mathbf{f}$ is a vector of all possible FDR-cutoffs sorted ascendingly, and $\mathbf{b}$ is a vector of the corresponding F-beta scores. $FDR_{(METASPACE\_thresh)}$ is one of the four possible METASPACE-default FDR thresholds (5%, 10%, 20%, and 50%).

## Training and cross-validating the model

We employed a ranking-based model using gradient boosting decision trees implemented using the CatBoost framework[10]. As input features, we used the original MSM features (spatial isotope, spectral isotope and spatial chaos) in addition to the newly introduced features (relative and absolute m/z error), five features in total. Two independent CatBoost models were built for animal and plant datasets independently. The models were first initialized using the "CatBoost" method using PairLogit as the loss function and fitted on 600 and 180 training datasets for 1000 iterations for animal and plant datasets, respectively. In each iteration, the decision trees are built in such a way that it improves the previous trees' output based on a loss function. For each dataset, a decision tree at a specific iteration scores all decoys and target ions based on their feature scores, using combinations of pairwise objects where one is considered a winner (target ion) and the other a loser (decoy ion). The pairlogit loss function[13] selects the best tree that maximizes the positive difference between the tree score for the target vs decoy ion. To evaluate the model and ensure that it was not overfitting, we performed cross-validation where the training datasets were split into 5 splits, where 80% of the dataset were used for training and the remaining 20% were used for evaluation (see Evaluation metrics) while maintaining the relative size of each context constant. The final prediction score of the model was scaled [0,1] using min-max scaling based on the leaf values of the decision trees so that scores closer to 1 denote high-confidence annotations and vice versa.

## Evaluation metrics and annotation database comparison

In order to evaluate the performance of METASPACE-ML compared to the rule-based approach, we compared how well their respective scores were able to rank target ions relative to decoy ions. We used Mean Average Precision (MAP), a commonly used metric that provides a comprehensive evaluation of the ranking accuracy and precision[14]. Precision is defined as the number of targets in the top "k" ions of a ranked list, divided by k. Then, the average precision for each ion in the ranked list is calculated followed by taking the mean of these average precision over all datasets. MAP scores were calculated for each cross-validated dataset for each of the 5 splits (see Training and cross-validation) and for the testing datasets.

While MAP and PR-AUC evaluate the ranking quality of META-SPACE-ML, they do not necessarily quantify the desired increase in the number of annotations. So, we calculated the relative fold change and

the log 10 difference in the number of target ions captured relative to the rule-based approach at specific FDR thresholds (5%, 10%, 20%, 50%). In addition, we calculated ion coverage for the testing datasets to determine the percentage of ions in a testing dataset that the model has encountered in any training dataset. Supplementary Note S3 provides detailed information on the calculation of coverage, and the scores are provided for all testing datasets in Supplementary Data 5.

## Selection of datasets for database comparison

In order to appropriately evaluate the performance of METASPACE-ML and ensure its ability to generalize to new unseen data aligning with the type of datasets represented in METASPACE and annotated with different databases, we have selected 389 datasets (Supplementary Data 6) in a context-independent manner having the same configuration as mentioned in Supplementary Note S1. All 389 datasets were annotated against 3 different databases: CoreMetabolome, SwissLipids(2018-02-02), and HMDB (version 4), and 206/389 were also annotated using KEGG (version 1).

## Separation of target-decoy and feature importance

An important aspect of the ideal annotation score is its ability to optimally differentiate between targets and decoys. In order to examine this separation per dataset, we calculated three original rule-based scores plus the new m/z error features for each ion and projected both target and decoy ions onto a two-dimensional space using UMAP implemented in the M3C R package[15] using the default configuration parameters except for *n_neighbors* which we set to 20 to align with the fixed decoy sampling size of 20 during FDR estimation. In addition, to evaluate which features are most important in driving the METASPACE-ML model's prediction, we used *SHAP* values from the Shap python package[16]. SHAP values quantify the contribution of each feature to the final prediction score for a given ion in each dataset. Given the additive property of the *SHAP* values, the contribution of each feature to the final prediction was taken as the ratio of the absolute *SHAP* value per feature to the sum of absolute *SHAP* values across all features for each ion. This was performed on 720 testing datasets, and the results were aggregated per dataset using the median of *SHAP* contribution and visualized as either a density heatmap from ComplexHeatmap R package[17] or a ridge plot from ggridge R package[18].

## Testing for difference in intensities of annotations

To further investigate the characteristics of the newly captured ions by the METASPACE-ML model, we compared the intensity distributions of those ions to the ones captured by both METASPACE-ML and the rule-based approach for animal datasets. Using METASPACE API (https://metaspace2020.readthedocs.io), we retrieved the ion images for each target ion in a given dataset and calculated the 99% percentile intensity across all pixels, followed by Log 10 transformation and taking the median across all ions captured at specific FDR threshold (default 10%). The distribution of the median-transformed intensities per dataset for each approach (METASPACE-ML and MSM) was compared using the Wilcoxon test. In addition, we used the *ggbetweenstats* function in the ggstatsplot R package[19] to perform the pairwise Mann-Whitney test between the intensity distributions of ions only captured by METASPACE-ML compared to either all ions captured by METASPACE-ML or only ions captured by the rule-based approach for a given dataset. *P*-values were adjusted using the Benjamini-Hochberg correction[20].

## Enrichment analysis

To learn more about the types of metabolites that were only picked up by METASPACE-ML, we performed a hypergeometric test to identify the molecular classes that were enriched in those metabolites.

Accordingly, we retrieved the class and subclass information for all annotated metabolites from the HMDB database (version 4) and used the HMDB "subclass" as background for enrichment. Then, for each dataset and subclass, we first filtered annotations with 10% FDR and then performed a two-tailed Fisher exact test where we considered the log fold enrichment as described in ref. 21 as a proxy for the enrichment score. Finally, we filtered significantly enriched terms (*p*-value < 0.05), and only terms that were enriched in at least 10% of the total number of input datasets were considered for visualization purposes. The complete enrichment results per dataset, context, and term can be found in (Supplementary Data 7).

## LC-MS/MS bulk validation of METASPACE-ML annotations

LC-MS/MS analysis was performed on an Agilent 1260 liquid chromatography (LC) system (Agilent, CA, USA) coupled to a Q Exactive Plus Orbitrap high-resolution mass spectrometer (Thermo Scientific, MA, USA) in positive ESI (electrospray ionization) mode.

Chromatographic separation was carried out on an Ascentis Express C18 column (Supelco, PA, USA; 100 × 2.1 mm; 2.7 μM) at a flow rate of 0.25 mL/min. The mobile phase consisted of water:ACN (40:60, v/v; mobile phase phase A) and IPA:ACN (9:1, v/v; mobile phase B), which were modified with a total buffer concentration of 10 mM ammonium formate + 0.1% formic acid. The following gradient was applied (min/%B): 0/10, 1/10, 5/50, 10/70, 18/97, 23/97, 24/10, 28/10. Column temperature was maintained at 25 °C, the autosampler was set to 4 °C and sample injection volume was 10 μL. Analytes were recorded via a full scan with a mass resolving power of 70,000 over a mass range from 150–900 *m/z*. MS/MS fragment spectra were acquired at 35,000 resolving power and stepped collision energies [%]: 10/20/30. Ion source parameters were set to the following values: spray voltage: 4000 V, sheath gas: 30 psi, auxiliary gas: 10 psi, ion transfer tube temperature: 280 °C, vaporizer temperature: 280 °C.

Data was processed using MS-DIAL 4.9.221218[22]. Feature identification was based on the MS-DIAL LipidBlast V68 library by matching accurate mass (*m/z* tolerance: 0.005), isotope pattern, and MS/MS fragmentation (*m/z* tolerance: 0.02) data (matching score threshold: 90%). To remove misannotations and to enhance confidence in lipid identification, intra-class elution patterns of lipid species were checked for consistency by relying on the expected chromatographic behavior on reversed-phase columns within homologous lipid series, considering carbon chain length and degree of saturation as the main factors[23].

For each of the 10 datasets (Supplementary Data 8) from[6], we matched the formula and adduct captured by either METASPACE-ML or the rule-based approach at 10% FDR with the LC-MS/MS matched formula + adduct. Only positive adducts (+ H, + Na, and + K) were considered. Accordingly, for each dataset, we got a 2 × 2 contingency table where true positives (TP) are defined as the set of ions that have < 10% FDR and bulk-validated, false positives (FP) are those that have < 10% FDR but not bulk validated, false negatives (FN) are those that have > 10% FDR and bulk-validated, and true negatives are those that > 10% FDR and not bulk validated. We calculated these metrics in either an exclusive manner for each approach (i.e., considering ions that are < 10% FDR for one approach but not the other) or inclusive. Finally, we report the common diagnostic metrics: true positive rate (TPR), false positive rate (FPR), and false negative rate (FNR) for each dataset.

## Hardware configuration for model training

Model training was executed on the EMBL in-house high-performance computing cluster. The training utilized AMD Epyc CPU nodes, each equipped with up to 128 cores, 256 threads, and a maximum of 384 GB of memory. Each training and evaluation job demanded a maximum of 50 and 30 GB of memory per core, and it was parallelized by splitting the job into 4 and 2 tasks, with each task employing 4 and 2 processors on the same node, respectively.

## Context Explorer Shiny app

To assist the user in the assessment of the reliability of prediction results per context and to provide more information about coverage, we developed the "METASPACE-ML: Context Explorer" web app (https://t.ly/q-nb5) using the R Shiny framework. The interactive web app allows the user to match the context closest to their dataset or just choose one or more contexts to further explore based on the 6 variables defined in (the context-dependent selection of training and testing datasets). Once the selected context(s) has been defined, the user will be able to view and download the corresponding training and testing datasets covering the selected context(s) and will be able to view all the evaluation and enrichment plots pertaining to datasets in such context(s). The app is only intended to view and interact with the results of the model on testing datasets in the specified context(s). It is not designed to predict outcomes for new datasets in similar contexts.

## Statistics and reproducibility

All statistical analyses were performed using R version 4.1.2 and Python version 3.8.13. The specific tests used are detailed within the respective figure legends. Unless otherwise stated, data are presented as the median with interquartile range. For comparisons between the two groups, a two-tailed Wilcoxon signed-rank test was used. For comparisons involving more than two groups, the Kruskal-Wallis test was employed. For repeated measurements, a paired Wilcoxon signed-rank test was applied. *P*-values were corrected for multiple comparisons using the Benjamini-Hochberg procedure unless stated otherwise. A *p*-value of less than 0.05 was considered statistically significant unless stated otherwise. No statistical method was used to pre-determine the sample size. No data were excluded from the analyses unless specified otherwise. The experiments were not randomized; The Investigators were not blinded to allocation during experiments and outcome assessment.

## Reporting summary

Further information on research design is available in the Nature Portfolio Reporting Summary linked to this article.

# Data availability

All datasets analyzed in this study are publicly available. The public training and testing datasets are available on METASPACE, and more information on each dataset can be accessed from Supplementary Data 5. Context-specific datasets can be accessed from the Shiny-based web app (https://t.ly/q-nb5). The LC-MS/MS data used for validation are publicly available at the MetaboLights repository under accession code MTBLS378. Raw source data is also available at the BioStudies repository under accession code S-BIAD1283.

# Code availability

The METASPACE-ML code for inference is available at https://github.com/metaspace2020/metaspace and covers scores calculation, METASPACE-ML, and FDR calculation based on the trained model. The METASPACE-ML models are available at https://github.com/metaspace2020/metaspace/tree/master/metaspace/scoring-models.

Codes and source data for generating the figures in this study are deposited at https://github.com/Bisho2122/METASPACE-ML_reproducibility, and the version of code used in this study is available via Zenodo with https://zenodo.org/doi/10.5281/zenodo.12798641.

Annotation using METASPACE-ML can be done directly in METASPACE during dataset submission in Annotation settings → Analysis version. Code to reproduce the curation of CoreMetabolome is available at https://github.com/DinosaurInSpace/core_metabolome. The source code for the Shiny web app is available at https://github.com/alexandrovteam/metaspace-ml-context-explorer.

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

## Acknowledgements

We acknowledge software support by a METASPACE software developer Lucas Maciel Vieira. We thank Tim Rose and Måns Ekelof for their time, consultation, and suggesting ideas to improve the analysis. The work was supported by grants from the European Research Council Horizon2020 CoG with grant agreement 773089 (T.A.), European Research Council HORIZON PoC with grant agreement 101101077 (T.A.), European Commission HORIZON with grant agreement 101092644, 101092646 (B.W., L.S., S.M., and T.A.), National Institutes of Health NIDDK KPMP2 (L.S., S.M., and T.A.) and NHLBI LungMAP2 (T.A.).

## Author contributions

B.W. has conceptualized substantial improvements of METASPACE-ML, trained the final METASPACE-ML model, expanded the validation methodology, performed data analysis and visualization, and wrote the manuscript. L.S. has conceived the METASPACE-ML approach, implemented it, and conceived and performed key steps of validation. C.M.R. has developed and curated the CoreMetabolome database. B.D. has performed re-analysis of LC-MS/MS data. S.M. has provided support on METASPACE and reprocessed datasets. T.A. has obtained funding, supervised the study, and edited the manuscript. All authors reviewed the manuscript.

## Funding

## Competing interests

T.A. holds patents on imaging mass spectrometry and leads a startup on single-cell metabolomics incubated at the BioInnovation Institute. The remaining authors declare no competing interests.
