## [Peer Review File · Nature Communications]

METASPACE-ML: Context-specific metabolite annotation for imaging mass spectrometry using machine learningEditorial Note: This manuscript has been previously reviewed at another journal that is not operating a transparent peer review scheme. This document only contains reviewer comments and rebuttal letters for versions considered at *Nature Communications*.

Reviewer #1 (Remarks to the Author):

The revised version of the manuscript by Wadie et al provided significant improvements over the 1st version. However, few points remind to be addressed before manuscript can be considered for the publication:

the core_metabolome generation code issues

The code for core_metabolome needs to be carefully updated, the current code will not run directly.

e.g,

```
https://github.com/DinosaurInSpace/core_metabolome/blob/master/core_metabolome_db.ipynb # cell # In [39]
```

there is an error that “core_metabolome” is not defined.

```
“”“
```

```
core_metabolome.to_pickle('core_metabolome_v1.pickle')
```

```
core_metabolome.to_csv('core_metabolome_v1.txt', sep='\t', index=False)
```

```
“”“
```

The author should update the code and allow other people to run the code.

Additionally, the pickle format should be changed to more accesable format like csv or json format and directly provide the .csv/.json in github so people can use it directly using other languages other than Python e.g, R or Java.

about the merging of DB isomers.

As I can see from “core_metabolome_v1.csv”

The following ones:

4015 HMDB0007982 PC(16:0/20:4(5Z,8Z,11Z,14Z)) C44H80NO8P InChI...

4016 HMDB0007983 PC(16:0/20:4(8Z,11Z,14Z,17Z)) C44H80NO8P InChI ...

Are isomers that MS can not distingusih, in version 3 with LMSD, there will be more isomers, the authors have to explain a bit more about the isomers in this DB.

For lipids there are C=C position isomers like above, and there are sn isomers e.g, PC(16:0/18:0) vs PC(18:0/16:0).

What I see from the current python code is that there is one value in the id column for the core_metabolome_v3.ipynb. However, some lipids/metabolites are presented in multiple databases, how the multiple IDs are stored is not very clear to me. Additionally, there are 7 databased mentioned in the generation of the core_metabolome database, however, I saw only IDs from KEGG, MSMLS, and HMDB. Either there is something wrong from the “core_metabolome_v1.csv” or the code. The authors should recheck the overall generation of the core_metabolome.

Decoy data for taining

If I understood correctly, the decoy dataset was created by generation of various adduct from the known elemental compositions from the database. However, when considering the real application to braoder range of datasets, e,g skin lipids, there are a list of omega-hydroxy Cer/SM that not list yet in LIPIDMAPS or HMDB, the issue of these un-trained lipids needs to be tested or with disclaimer to the users that they have to retrain the model if the required search space is epected to exceed the coverage of CoreMetabolites database.

Compound annotation remarks

As I understood, the algorithm direct output would be the elemental composition rather than

the metabolite annotation.

(<https://metaspace2020.readthedocs.io/en/latest/content/examples/colocalized-annotations.html>) The elemental composition range has been broadened from single database to the CoreMetabolites database, however, the multiple isomeric lipids/metabolites annotations to the identified elemental composition is still not very clear for me. A given example with the interface and direct compare of the MSI image would help to clarify the final outcome and the annotation of the metabolite elemental composition.

It would be more straightforward, if the authors can provide one example in the positive mode and one example in the negative mode for side by side comparison of a real example of True positive metabolite with its score and the scores for the corresponding 20 decoy. Also a False positive of a metabolite with corresponding target metabolite score. This will give non-bioinformatic researcher more idea about the score.

Reviewer #2 (Remarks to the Author):

I agree with the authors that molecular annotation of metabolites in spatial metabolomics experiments using mass spectrometry imaging is a serious challenge. I furthermore applaud the authors for providing a very thorough examination of the prediction results and offering various viewpoints on the results, providing a perspective on the capabilities of METASPACE-ML that surpasses what was offered in the previous version of this manuscript. The substantial increase in training data over the previous version, is without doubt an important advancement to the capabilities of the model.

However, several important aspects that were raised in the previous version of manuscript seem to remain insufficiently resolved and continue to substantially impact the METASPACE-ML model's applicability and significance. Two major comments:

(1) Limited coverage of the spatial metabolomics space in the training data. METASPACE-ML is trained on a relatively narrow set of MSI experiment types and tissue types, substantially impacting its applicability and reliability outside those data types. This is at tension with the title, abstract, and introduction suggesting a rather broad coverage. I understand the developers have increased the number of training datasets to address this criticism, but if the added datasets do not provide broader and more varied coverage, the additional training only increases the model's performance for the data types trained on while the width of coverage of the spatial metabolomics space is not automatically increased.

(2) Contexts and reliability of METASPACE-ML predictions. Whether METASPACE-ML predictions on a new dataset (outside the training and testing data) are reliable seems to be linked to whether the new dataset fits into a context that was trained on. However, the assessment of whether a new dataset supplied by a user is sufficiently close to a context that was trained on seems to be passed on to the user, who is not well-equipped to make that call. The assessment is muddled further by an unclear definition of "context".

These issues narrow METASPACE-ML's scope and applicability, and impact its reliability for a substantial part of the spatial metabolomics space (e.g TOF-based MSI datasets). The manuscript seems to acknowledge these issues, and states in the Discussion "With the growing number of public imaging MS datasets available in METASPACE, we can train future generations of METASPACE-ML by including more datasets and covering more contexts." However, as stated previously, that does not resolve the issue at this time and substantially impacts the current use and significance of the model presented in this manuscript.

[Limited coverage of the spatial metabolomics space in the training data]

"we selected substantially more datasets for training and testing"

--- Adding numbers is not sufficient to address the experiment/sample variety and imbalance in the training data. For example, if the additional datasets do not cover more instrumentation and samples variety or still have an imbalanced composition, little progress is made towards resolving these issues.

"We introduced two METASPACE-ML models, for animal and plant datasets, respectively"

--- Similar remark as above. Building a separate model for animals does not increase the width of coverage of animal species, it allows the model to use more of its capacity towards the few species that were represented in the training data.

"adopted a more balanced and representative approach for dataset selection and substantially expanded the pool of datasets for both training and testing datasets"

--- Adopting a "more balanced" approach is not the same as adopting a balanced approach. I understand that adding more datasets can increase the coverage and balance of the model training process, but it doesn't have to, and it does not mean the model is sufficiently general to give good results in a general spatial metabolomics IMS case. The representation and balance is shown in e.g. Figure 2C and 2D, where certain types of experiments are clearly in the majority and others in the minority.

"We have addressed this concern by expanding our evaluation to include a more diverse set of datasets which even extend beyond the distribution of the training data."

--- Expanding the number of datasets trained on does not automatically ensure that a new external dataset presented by a user falls within the distribution that was trained on, regardless of how large the training data set is. The manuscript needs a more thorough assessment of applicability and reliability when METASPACE-ML is presented with an unseen dataset. At what point should a user feel confident about the predictions METASPACE-ML delivers for their own dataset? Please provide guidance.

"The omission was a result of insufficient datasets to ensure the statistical reliability of the evaluation results. However, as more TOF datasets become publicly available, we anticipate a more comprehensive demonstration of the METASPACE-ML model's capabilities."

--- I appreciate that there might not be sufficient TOF datasets in METASPACE to train on. However, please then narrow the scope of the applicability of the model in the title, abstract,

and description to reflect the instrumental subareas where it can provide good predictions. Currently, the manuscript suggests a broader scope than can be delivered by the datasets used in the training. Try to make precise calls on where the model is to be trusted and acknowledge more explicitly where the model is not reliable.

"offering improved accuracy and sensitivity compared to the current state-of-the art methods"
--- Why methods plural? Which methods do the authors mean here? The rebuttal states "we have opted not to include a comparative evaluation of multiple algorithms in the current manuscript" and "our primary focus is on presenting the development and evaluation of METASPACE-ML with Gradient Boosting Decision Trees compared to its rule-based predecessor", implying and confirming that the manuscript carries a comparison only to one method, namely the previous METASPACE approach using the MSM rule. Please avoid such broad statements in the abstract and give a precise account of what is supported by the manuscript that follows.

"We developed an advanced strategy for selecting training and testing datasets which have high quality and in a balanced way represent different contexts of technology and sample types."
--- If the model is trained on a subset of datasets that is labeled as "high quality", are the model's predictions not skewed towards high quality datasets? What happens if a "low quality" dataset is presented to the model? Can we detect that scenario? More importantly, how does a user of METASPACE-ML assess whether their own dataset fits the model's definition of "high quality"?

"We trained METASPACE-ML on 600 and 180 animal and plant datasets, and evaluated on 930 datasets, all submitted by 47 labs."

--- The number of datasets used in the training phase does not automatically address the issue of coverage and whether we can trust the predictions for a user's dataset. If the training data is not representative for the user's dataset and the testing data is neither, the performance on the training and testing dataset does not help much in making the model perform well for that user's dataset. This issue happens in many machine learning applications, and it should be explicitly acknowledged in the manuscript to help guide the reader.

"This warrants further work for evaluating METASPACE-ML for less common data e.g. from cultured cells in single-cell metabolomics, or data from industrial labs, where the ability to deposit data publically may be limited."

--- I agree with this statement. Please narrow the scope of the applicability claims made in the title, abstract, and introduction to reflect this.

"With the growing number of public imaging MS datasets available in METASPACE, we can train future generations of METASPACE-ML by including more datasets and covering more contexts. Importantly, we envision training custom METASPACE-ML models for specific contexts or data providers, thus achieving the best metabolite annotation for their data."

--- I agree with the authors. However, that is not what is currently being offered by the METASPACE-ML model. Most users will be looking to apply the currently available model described in this manuscript to their own data, similar to how the MSM approach has been used. How can they assess whether they receive reliable and relevant results when they do that?

"Our study represents a major advancement in metabolite annotation for spatial metabolomics by introducing the first machine learning model trained and evaluated on a large collection of diverse imaging MS datasets that consistently outperforms the rule-based approach."

--- The manuscript reports the diversity of the training data that was used and, as the manuscript indicates, the coverage of the space of potential MSI experiments is rather limited. For example, TOF analyzers, probably one of the most common mass analyzer types for MSI experiments, is not (well) represented in the training data. METASPACE-ML seems quite good in the MSI data types on which it is trained, but given that it has (nearly) only seen Orbitrap and FTICR training examples, it does not seem suited for use on TOF-based MSI datasets. Does this not narrow the scope of applicability for METASPACE-ML? This point is e.g. relevant to METASPACE-ML's application to lipid annotation, since it is quite common for lipid MSI datasets to be acquired on a TOF instrument.

If the argument is that spatial metabolomics is done mostly with Orbitraps and FTICRs (I'm not sure that statement would hold, but let us assume), then the argument only further underlines the narrow scope of where METASPACE-ML can be reliably applied.

Please treat the point of training and applicability of the model more prominently in the manuscript.

[Contexts and reliability of METASPACE-ML predictions]

"we considered technology-biology configurations or contexts relevant for practitioners in the imaging mass spectrometry field"

--- What constitutes a technology-biology configuration? Does this mean a combination of the six properties available for each dataset? Can it be a more broad combination of only five of the six properties? Please be precise in the definition of a context. Also, please specify how a practitioner in the mass spectrometry imaging field should use these contexts? At what point is the user's dataset a match for a specific context? When all six properties match? When four of the six properties match? Please elaborate.

Figure 2C and 2D

--- Why is Figure 2C's training data missing the "mz_class" dataset parameter while Figure 2D's test data includes it?

"the balanced representation of MALDI vs. DESI-based human datasets in our evaluation set enables a fair comparison"

--- What do the authors mean by fair here? Figures 2C and 2D show MALDI dominating the DESI representation, also in the human subset. Please explain how to get to this statement on fairness.

"we believe that this contextualization allows for an unbiased evaluation of METASPACE-ML's performance within each specific context, without compromising its assessment in other contexts."

--- If the contexts only play a role in the evaluation of the model post-training, how do they affect the issue of coverage, applicability, and balance of the trained model? The sampling procedure explained in the Methods section of the manuscript increases the coverage within the datatypes

already used for training, but it does not increase the coverage towards more data types than the ones already considered. Furthermore, even if performance within a context is high quality, how does one find out that their own dataset is adequately covered by the datasets captured by that context?

"We have now provided a more detailed assessment of the evaluation results per context which stratifies the results based on polarity ,organism, sample-type, analyzer, ionization source, and molecular species (Lipids, small-molecules, both)."

--- Same question as for the previous remark.

"we have evaluated reliability of the model per context to ensure fair and unbiased comparisons."

--- Same question as for the previous remark.

"with the introduced contextualization of the results, the user will be able to align their metadata with the ones defined by the context."

--- How does a user align their own data with a context as described in the manuscript? At what point is the metadata a match? Is a fit of the analyzer type sufficient? Or should it be more precise? Please provide clear guidance for the reader on the alignment process.

"a balanced representation of various spatial metabolomics contexts"

--- Which contexts specifically? Please narrow the scope of the abstract and title to reflect the contexts where METASPACE-ML would be expected to do a good job.

"In summary, the balanced and context-specific selection of datasets helped us select a large representative set of datasets and will provide the end-users the granularity of the represented sample types and protocols to judge on the reliability of the model predictions."

--- This statement suggests that the responsibility for assessing whether prediction results are reliable for a user-provided dataset lies with the user. On the basis of what information can the user assess this? First, at what point should a user consider their dataset as being matched by a context? Please provide guidance. Second, even if a user's dataset can be assigned as being a member of a particular context, that does not mean that the training datasets employed in that context are relevant or representative for the content of the user's dataset. It is very good that the authors are adding datasets compared to a previous iteration of this manuscript. However, this does not address the issue of whether METASPACE-ML predictions are reliable for a user's unseen dataset.

"each with their unique combination of contexts encompassing both analytical and biological metadata to provide a more granular approach for end-users to align and cross-reference their datasets with the training and evaluation datasets"

--- How should users align and cross-reference their datasets with the datasets used to build the model? If the argument is that users should see if one of the contexts examined in the paper is similar to the context of their own data, there are still two issues preventing a user from being able to make that assessment:

1) At what point is a training context sufficiently close to the user's dataset? What if different contexts could potentially fit the user's dataset and the performance for these contexts is quite different? Is it sufficient to match on the analyzer, or does the user's dataset need to match several meta-parameters? How is the user equipped to make this assessment? The experts on

the model, i.e. the authors, seem better equipped to make that call?

2) Even if a user's dataset can be correctly aligned with a specific trained context, how can the user assess whether that context has seen examples relevant to the dataset of the user (enabling good prediction)? Stratifying the dataset database into specific contexts does not solve the fundamental problem of whether the model has seen examples relevant to the new dataset, i.e. whether the user's dataset fits the trained distribution. It is not clear from the manuscript how the user can assess whether the results are reliable for their own dataset. The argument in the manuscript seems to be that the model has been trained on the largest collection of MSI datasets available. However, that is not the same as the model being trained on a collection of MSI datasets that is sufficiently large, and the latter is necessary to ensure reliability in general, a claim suggested by the title, abstract, and introduction of the manuscript. If the training data is not sufficiently large, as seems to be the case here, the alternative is to assess whether a dataset one wants to predict for matches the training data sufficiently, but this does not seem to be provided here.

"The datasets were classified based on possible combinations of the following 6 variables : polarity, ionization source, mass analyzer, mz class, sample type and species."

--- Which contexts were considered exactly? All possible combinations of these six variables?

"for some models, certain contexts are not represented due to insufficiency of available datasets to cover both testing and training."

--- Which models saw which contexts? Please specify.

"final list of selected datasets for each context and kingdom can be found in Supplementary Table 5."

--- The kingdom is indicated in Table S5, but I don't seem to be able to find an easy mapping to a context. Which contexts (or parameter combinations) were actually trained? Are these all possible combinations of the 6 variables, or less? If I missed it, please point me to a definitive list of the contexts. If not, please provide a list and their connection to the models.

"while maintaining the relative size of each context constant."

--- Please elaborate. Two independent CatBoost models were built for animal and plant datasets independently, but which contexts play a role in the training phase is not very clear. Please provide an exact record of the training of each of the two models. Please provide an exhaustive list of the contexts used. Since the contexts are central to the improvements made to the previous version of this manuscript, it is important to give a strong and rigorous treatment of this aspect.

Figure S1 and S2

--- Please provide a clear definition and enumeration of what constitutes a "context". Figure S1 mentions six contexts: polarity, source, analyzer, mz_class, sample_type, and species. Figure S2, however, mentions that there are 24 contexts in certain situations (along the vertical axis labeled "number of contexts"). In Figure S4, it is mentioned that "A flow from the first to the last node represents a single context", but in that figure "mz_class" is not part of the Sankey diagram, so does that mean that a context is not specific to "mz_class"? That seems in contradiction with Figure S2. Also, the drawing in Figure S4 suggests that there are 32 different contexts, which does not line up with statements elsewhere. Then again, Figure S8, shows 36 contexts based on the labels polarity, species, sample_type, but it takes source+analyzer

together as one and at the same time seems to leave `mz_class` out of the specification. Also in Figure S8, 13 of the 36 contexts seem to be empty, presumably due to not enough datasets in those label combinations? Does this mean there are actually 23 non-zero dataset contexts? If so, does that not mean the training data is still unbalanced? The matter is confused further by Figure 4C in the main manuscript, suggesting there are 24 contexts in total in the animal-based testing datasets, and Figure S10 suggests there are 7 contexts in total in the plant-based testing datasets. While one could consider Figure 4C to line up with Figure S1, their total context number differs from Figure S11, which lines up with S8 if we deduct the empty boxes. To avoid these ambiguities, please be precise in definitions and rigorous in wording throughout. Furthermore, if the user is to assess whether METASPACE-ML is reliable for their own dataset and there's only one model for several contexts, does the unbalanced distribution of dataset numbers across different contexts not mean that the model would be skewed towards the majority meta-information labels in the total training data? How does a user know whether their own dataset fits sufficiently with the majority representatives that the final model was trained on?

The manuscript can use further elaboration on what constitutes a context exactly, what is an exhaustive list of the contexts considered, and how is the interaction between contexts and any other steps performed, e.g. cross-validation?

[Other remarks]

My original comment was "When a dataset's m/z range is labeled as 'high', does this mean that its m/z range goes higher than m/z 500? How high? What is the maximum m/z measured? Please report a summary of m/z ranges covered." In response, a new sentence in the manuscript reads as follows "Finally, datasets were also classified based on their m/z range, where datasets having maximum m/z ≤ 400 are classified as small molecules and those having minimum m/z > 500 are classified as Lipids. The rest of datasets with min m/z < 500 and max m/z > 400 are classified as "Lipids and small molecules"

--- This change does not answer my original question. Please specify the maximum m/z of the ranges. For example, for the "min m/z < 500 " datasets, what is the extent (in m/z units or Th) of their ranges covered? Up to what m/z value do we consider molecular species in METASPACE-ML, and why is that an appropriate range?

"we have validated some of the newly captured annotations by METASPACE-ML compared to the rule-based approach by performing LC-MS/MS bulk analysis of mouse brain datasets from our previous paper (ref)"

--- Which paper is being referred to here?

Figure 2C-D "Sankey diagrams showing the breakdown of training (C) and testing (D) animal datasets by different parameters."

--- The training and testing datasets should have an identical distribution. However, the Sankey diagrams show differences in distribution, for example on the top right of the diagrams around the "whole organism" label. Why is there a difference in distribution present?

"The mean average precision (MAP) was used to evaluate the quality of target vs decoy ion

rankings."

--- Many different measures could be used to evaluate performance here. Each measure has its own sensitivities and blind spots. Please elaborate why MAP is the right measure to use here.

"A dot represents a dataset and y-axis represents the Log10 absolute difference where negative values are those where MSM had more annotations than METASPACE-ML and vice-versa."

--- There are quite some dots on the negative side, suggesting that for those datasets MSM finds certain annotations that are not found by METASPACE-ML. Please elaborate more on the nature of those annotations.

"our approach has the potential to enhance imaging mass spectrometry and spatial metabolomics, and to have far-reaching implications for biology, medicine, and pharmacology."

--- Please avoid these rather broad statements and instead provide the exact areas within which reliability of the model is high.

"based on their m/z range"

--- Please use "m/z" in the text of the manuscript. "mz" is incorrect.

"30 testing datasets per context were selected first for each kingdom"

--- Why 30? Why is this number a reasonable value to use?

Equation 3 seems to imply that the difference between the theoretical m/z value of the first isotope and the observed mean m/z value of the first isotope can never exceed 1. Is that the case? Is this enforced beforehand? If so, I don't think I saw it in the preceding method description. If not, what happens if the m/z difference is larger than one?

Equation 4

--- Same questions as for Eq. 3. in the previous remark.

"we choose beta = 0.5 to put more weight on the precision"

--- Why is 0.5 an appropriate parameter value for beta?

[Typos and smaller notes]

"mz <= 400"

--- The property mentioned here is the mass-to-charge ratio and should therefore be written as "m/z". The authors use "mz", which (incorrectly) implies mass multiplied with charge to be the property. Please be exact in notation.

"elch"

"we can calculate the proportion of overlapping intervals after shifting m/z values by a given ppm"

--- Equation (1) does not seem to be shifting the positions of m_i and m_{i+1} , but rather checking the distance.

Table S5

--- This table seems to use "Lipids_and_small_moleclues" throughout. Please correct the spelling.

Reviewer #3 (Remarks to the Author):

The authors adequately addressed my comments.

Summary of changes

We thank the reviewers for providing additional comments to improve the manuscript.

We have addressed all the comments accordingly and made several changes.

Firstly, we updated the code to reproduce the core_metabolome database, which can be accessed at https://github.com/DinosaurInSpace/core_metabolome.

We also rephrased the main text to emphasize "representativeness" over "balanced" to provide a more accurate description of dataset selection. Additionally, we appended and modified the abstract, introduction, and discussion to explicitly define the scope where METASPACE-ML can be applied. To better define contexts, we added explanatory statements and modified supplementary tables to explicitly list all contexts mentioned in this study. We fixed typos and modified main and supplementary figures to include m/z_class as requested. Furthermore, we added a note on testing dataset coverage in the supplementary information (Section 3).

To address the major comments regarding the alignment of a new dataset to defined contexts, we developed a web app in the form of a ShinyApp (<https://t.ly/q-nb5>) to help users align their datasets against defined contexts and check the model performance on datasets that are similar based on context definition. Using this app, users can get more information about which datasets the model was trained and tested on, as well as evaluate the performance of the model on those datasets.

Responding to Reviewer #1

Comment 1: *The code for core_metabolome needs to be carefully updated, the current code will not run directly.*

e.g,

https://github.com/DinosaurInSpace/core_metabolome/blob/master/core_metabolome_db.ipynb

there is an error that "core_metabolome" is not defined.

"""

```
core_metabolome.to_pickle('core_metabolome_v1.pickle')
```

```
core_metabolome.to_csv('core_metabolome_v1.txt', sep='\t', index=False)
```

"""

The author should update the code and allow other people to run the code.

Additionally, the pickle format should be changed to a more accessible format like csv or json format and directly provide the .csv/.json in github so people can use it directly using other languages other than Python e.g, R or Java.

Response 1:

We thank the reviewer for spotting it and we apologize for the oversight.

Action taken:

We have changed the code and made it more reproducible by adding the required data to reproduce `core_metabolome_v1`. We also replaced any `.pkl` format to `.csv` for better interoperability as requested. The new updates were merged in the same github repo (https://github.com/DinosaurInSpace/core_metabolome).

Comment 2: # about the merging of DB isomers.

As I can see from “`core_metabolome_v1.csv`”

The following ones:

4015 HMDB0007982 PC(16:0/20:4(5Z,8Z,11Z,14Z)) C44H80NO8P InChI...

4016 HMDB0007983 PC(16:0/20:4(8Z,11Z,14Z,17Z)) C44H80NO8P InChI ...

Are isomers that MS can not distinguish, in version 3 with LMSD, there will be more isomers, the authors have to explain a bit more about the isomers in this DB.

For lipids there are C=C position isomers like above, and there are sn isomers e.g, PC(16:0/18:0) vs PC(18:0/16:0).

What I see from the current python code is that there is one value in the id column for the `core_metabolome_v3.ipynb`. However, some lipids/metabolites are presented in multiple databases, how the multiple IDs are stored is not very clear to me. Additionally, there are 7 databases mentioned in the generation of the `core_metabolome` database, however, I saw only IDs from KEGG, MSMLS, and HMDB. Either there is something wrong from the “`core_metabolome_v1.csv`” or the code. The authors should recheck the overall generation of the `core_metabolome`.

Response 2:

We agree that the isomers listed are indeed indistinguishable by MS, however the purpose of using LipidMaps in this curation was rather to whitelist HMDB isomers that should be included in CoreMetabolome.

During the annotation process, METASPACE considers sum formulas, yet preserving information which molecules the formulas correspond to. In the web app or exported annotation results, METASPACE shows all isomers (and also isobars) which can correspond to a particular ion, and warns users about their presence.

We would like to emphasise that the discussed isomeric (or isobaric) ambiguity is not a limitation of METASPACE or METASPACE-ML but a general limitation of imaging mass spectrometry technology.

Action taken :

Same as in Response 1, we have updated the code for reproducibility of constructing the CoreMetabolome database and included an explanation for all the databases used for creating CoreMetabolome and how they were processed.

Comment 3: # Decoy data for training

If I understood correctly, the decoy dataset was created by generation of various adduct from the known elemental compositions from the database. However, when considering the real application to broader range of datasets, e.g skin lipids, there are a list of omega-hydroxy Cer/SM that not list yet in LIPIDMAPS or HMDB, the issue of these un-trained lipids needs to be tested or with disclaimer to the users that they have to retrain the model if the required search space is expected to exceed the coverage of CoreMetabolites database.

Response 3:

We would like to clarify that the decoy adducts are not generated from the elemental composition of the database, rather as explained in our previous paper (Palmer et al, Nature Methods 2017). For each sum formula from the molecular database of interest, and for each ion adduct requested by the user, an implausible elemental adduct (decoy adduct) was randomly chosen from the list of implausible adducts proposed by us e.g. +Be.

So the decoy adduct generation is completely independent of the selected annotation database. However, we do acknowledge in the discussion how the decoy generation could affect the model's performance and discuss potential improvements of the adduct selection process.

Comment 4: # Compound annotation remarks

As I understood, the algorithm direct output would be the elemental composition rather than the metabolite annotation. (<https://metaspace2020.readthedocs.io/en/latest/content/examples/colocalized-annotations.html>) The elemental composition range has been broadened from single database to the CoreMetabolites database, however, the multiple isomeric lipids/metabolites annotations to the identified elemental composition is still not very clear for me. A given example with the interface and direct comparison of the MSI image would help to clarify the final outcome and the annotation of the metabolite elemental composition.

It would be more straightforward, if the authors can provide one example in the positive mode and one example in the negative mode for side by side comparison of a real example of True positive metabolite with its score and the scores for the corresponding 20 decoy. Also a False positive of a metabolite with corresponding target metabolite score. This will give non-bioinformatic researchers more ideas about the score.

Response 4:

We would like to clarify that METASPACE-ML is part of METASPACE, so the annotation results on METASPACE webapp will also be in the same format.

As an example, please check out the following mouse brain dataset (https://metaspace2020.eu/annotations?db_id=22&ds=2024-05-02_17h21m34s). On the left, the annotation table shows the ions based on the sum formula and if you hover over a specific

annotation, a pop up will appear with molecule names and numbers of isomers and isobars. On the right, you can see the ion image of the selected ion, and below there are multiple tabs providing extra information as follows :

- Molecules tab shows the possible list of isomers (and potential isobars with a warning sign) based on the selected annotation database (in this case HMDB-v4).
- Colocalized annotations : shows ion images of co-localized ions.
- Diagnostics :
 - METASPACE-ML score and corresponding feature scores
 - Ion images for the top 4 isotopic peaks of that ion
 - m/z vs relative intensity of those 4 isotopic peaks showing both observed and theoretical intensities.
- Metadata: Dataset metadata

Regarding an example for target and decoy data, we would like to clarify that not all 20 decoys selected for a given ion can be found in the dataset. Accordingly, when estimating the FDR, only the decoys having a non-zero MSM annotation are considered in the ranking. But, the FDR is weighted to consider the 20 decoy sample size (**Check FDR estimation**) in the methods section for more details.

Action taken :

We are providing a supplementary table (Table S10) that presents the scores for both MSM and METASPACE-ML for both targets and decoys, along with their corresponding FDRs. For a given FDR threshold, such as 10%, a true positive is defined as a target ion with an FDR < 10%, while a false positive is a decoy ion with an FDR < 10%. This table includes the input data used to estimate the FDR for a specific group (+Na adducts) within a particular dataset (https://metaspace2020.eu/dataset/2018-12-14_16h34m31s).

Responding to Reviewer # 2

Summary of major comments

1. *Limited coverage of the spatial metabolomics space in the training data. METASPACE-ML is trained on a relatively narrow set of MSI experiment types and tissue types, substantially impacting its applicability and reliability outside those data types. This is at tension with the title, abstract, and introduction suggesting a rather broad coverage. I understand the developers have increased the number of training datasets to address this criticism, but if the added datasets do not provide broader and more varied coverage, the additional training only increases the model's performance for the data types trained on while the width of coverage of the spatial metabolomics space is not automatically increased. (Comments 1-11).*
2. *Contexts and reliability of METASPACE-ML predictions. Whether METASPACE-ML predictions on a new dataset (outside the training and testing data) are reliable seems to be linked to whether the new dataset fits into a context that was trained on. However, the assessment of whether a new dataset supplied by a user is sufficiently close to a context that was trained on seems to be passed on to the user, who is not well-equipped to make that call. The assessment is muddied further by an unclear definition of "context". (Comments 12-22)*

These issues narrow METASPACE-ML's scope and applicability, and impact its reliability for a substantial part of the spatial metabolomics space (e.g TOF-based MSI datasets). The manuscript seems to acknowledge these issues, and states in the Discussion "With the growing number of public imaging MS datasets available in METASPACE, we can train future generations of METASPACE-ML by including more datasets and covering more contexts." However, as stated previously, that does not resolve the issue at this time and substantially impacts the current use and significance of the model presented in this manuscript.

Actual comments

Comment 1: *"we selected substantially more datasets for training and testing"*

--- Adding numbers is not sufficient to address the experiment/sample variety and imbalance in the training data. For example, if the additional datasets do not cover more instrumentation and samples variety or still have an imbalanced composition, little progress is made towards resolving these issues.

Response 1:

We agree that adding more datasets doesn't necessarily imply increased diversity, representation or even model performance.

However, we would like to clarify that we don't claim that adding more datasets is a solution to improve representation and diversity. Rather, we claim that adding more datasets compared to the original submission allowed us to cover more contexts derived from public datasets in METASPACE. Having more datasets for each context exposes the model to different annotations from similar datasets, thus making the model less biased and improves the statistical reliability when it comes to evaluation of model performance.

Action taken:

We rephrased the text by emphasizing representativeness and avoiding the use of the word "balanced". We also changed the abstract to highlight that datasets selected are from the METASPACE knowledge base.

Comment 2: *"We introduced two METASPACE-ML models, for animal and plant datasets, respectively"*

--- Similar remark as above. Building a separate model for animals does not increase the width of coverage of animal species, it allows the model to use more of its capacity towards the few species that were represented in the training data.

Response 2:

We agree that having a separate model for animals does not increase the width of coverage of animal species.

However, we would like to clarify that the aim behind having two separate models is to contextualize the annotations exposed to the model with the assumption that some ions observed in plants are not observed in animal tissues and vice versa. Therefore, despite the imbalance in the types of species/organisms covered in METASPACE, it's still representative to its public domain.

Action taken:

Same as in Response 1.

Comment 3: *"adopted a more balanced and representative approach for dataset selection and substantially expanded the pool of datasets for both training and testing datasets"*

--- Adopting a "more balanced" approach is not the same as adopting a balanced approach. I understand that adding more datasets can increase the coverage and balance of the model training process, but it doesn't have to, and it does not mean the model is sufficiently general to give good results in a general spatial metabolomics IMS case. The representation and balance is shown in e.g. Figure 2C and 2D, where certain types of experiments are clearly in the majority and others in the minority.

Response 3:

We agree with the reviewer and we have rephrased the text accordingly.

Action taken:

Similar to Response 1, we have removed all instances of the word "balanced" and emphasized representation instead.

Comment 4: *"We have addressed this concern by expanding our evaluation to include a more diverse set of datasets which even extend beyond the distribution of the training data."*

--- Expanding the number of datasets trained on does not automatically ensure that a new external dataset presented by a user falls within the distribution that was trained on, regardless of how large the training data set is. The manuscript needs a more thorough assessment of applicability and reliability when METASPACE-ML is presented with an unseen dataset. At what point should a user feel confident about the predictions METASPACE-ML delivers for their own dataset? Please provide guidance.

Response 4:

We would like to clarify that our model is indeed evaluated on unseen test datasets. As mentioned in the previous responses, we don't claim that adding more datasets would increase representativeness. And when it comes to exposing the model to unseen datasets, that's what the testing datasets are for and we have evaluated the model on many different "unseen" datasets, even from contexts that the model wasn't trained on, and we show an unbiased evaluation of when the model was reliable and when it's not.

Action taken:

We have developed a new ShinyApp – a web app to help users check the model performance on datasets that are similar based on context definition. Using this app, users can get more information about which datasets the model was trained and tested on, as well as being able to evaluate the performance of the model on those datasets.

Comment 5: *"The omission was a result of insufficient datasets to ensure the statistical reliability of the evaluation results. However, as more TOF datasets become publicly available, we anticipate a more comprehensive demonstration of the METASPACE-ML model's capabilities."*

--- I appreciate that there might not be sufficient TOF datasets in METASPACE to train on. However, please then narrow the scope of the applicability of the model in the title, abstract, and description to reflect the

instrumental subareas where it can provide good predictions. Currently, the manuscript suggests a broader scope than can be delivered by the datasets used in the training. Try to make precise calls on where the model is to be trusted and acknowledge more explicitly where the model is not reliable.

Response 5:

We agree with the reviewer that it is imperative to make precise calls on where the model is to be trusted and explicitly report where it might not be reliable.

Action taken:

We added a paragraph in the discussion to explicitly mention the lack of TOF-based datasets in both training and testing sets. We also rephrased the text in abstract and introduction to better describe the scope of METASPACE-ML.

Similar to our Response 4, we would like to highlight that we have developed a new ShinyApp, a web app providing statistical results for each of the contexts that we used for training and testing, so a user can obtain context-specific results about their own context of interest.

Comment 6: *"offering improved accuracy and sensitivity compared to the current state-of-the art methods"*

--- Why methods plural? Which methods do the authors mean here? The rebuttal states "we have opted not to include a comparative evaluation of multiple algorithms in the current manuscript" and "our primary focus is on presenting the development and evaluation of METASPACE-ML with Gradient Boosting Decision Trees compared to its rule-based predecessor", implying and confirming that the manuscript carries a comparison only to one method, namely the previous METASPACE approach using the MSM rule. Please avoid such broad statements in the abstract and give a precise account of what is supported by the manuscript that follows.

Response 6:

We agree with the reviewer and apologise for the oversight.

Action taken:

We removed the statement from the abstract and rephrased accordingly to refer to the rule-based approach.

Comment 7: *"We developed an advanced strategy for selecting training and testing datasets which have high quality and in a balanced way represent different contexts of technology and sample types."*

--- If the model is trained on a subset of datasets that is labeled as "high quality", are the model's predictions not skewed towards high quality datasets? What happens if a "low quality" dataset is presented to the model? Can we detect that scenario? More importantly, how does a user of METASPACE-ML assess whether their own dataset fits the model's definition of "high quality"?

Response 7:

We would like to clarify the following points :

1. Regarding skewed predictions :

The criteria we used in the manuscript that we explained in the method section “Exclusion of low-quality datasets” are exclusion criteria to filter out datasets that can compromise the evaluation of the model. Consequently, the model is not inherently biased towards high-quality datasets but is designed to perform reliably across various dataset qualities that meet such minimum standards.

2. Handling Low-Quality Datasets and user assessment of dataset quality:

If a dataset that does not meet our exclusion criteria is submitted to METASPACE, it will be flagged, and the user will be notified of the potential issues. In such cases, we will recommend not relying on the model’s predictions and reprocessing the dataset before re-submitting it to METASPACE.

Action taken:

We have added the following sentence in the discussion “Newly submitted datasets that do not adhere to the defined quality criteria will be flagged and the corresponding users will be notified prior to processing”.

Comment 8: *"We trained METASPACE-ML on 600 and 180 animal and plant datasets, and evaluated on 930 datasets, all submitted by 47 labs."*

--- The number of datasets used in the training phase does not automatically address the issue of coverage and whether we can trust the predictions for a user's dataset. If the training data is not representative for the user's dataset and the testing data is neither, the performance on the training and testing dataset does not help much in making the model perform well for that user's dataset. This issue happens in many machine learning applications, and it should be explicitly acknowledged in the manuscript to help guide the reader.

Response 8:

We acknowledge that the users should be fully informed about the coverage of the training and testing data used for machine learning.

Specifically, we agree that it’s important for a user to understand whether their context is represented in the training and testing data. In order to address this, we have implemented a new webapp (ShinyApp) which, for a context of interest, shows to a user whether the datasets from this context were included in the training and testing sets (and the number of datasets from this context) and where were the statistical evaluation results.

Action taken:

We have developed a new ShinyApp – a web app to help users check the model performance on datasets that are similar based on context definition. Using this app, users can get more information about which datasets the model was trained and tested on, as well as being able to evaluate the performance of the model on those datasets.

Comment 9: *"This warrants further work for evaluating METASPACE-ML for less common data e.g. from cultured cells in single-cell metabolomics, or data from industrial labs, where the ability to deposit data publically may be limited."*

--- I agree with this statement. Please narrow the scope of the applicability claims made in the title, abstract, and introduction to reflect this.

Response 9:

We agree with the reviewer.

Action taken:

Same as Response 5.

Comment 10: *"With the growing number of public imaging MS datasets available in METASPACE, we can train future generations of METASPACE-ML by including more datasets and covering more contexts. Importantly, we envision training custom METASPACE-ML models for specific contexts or data providers, thus achieving the best metabolite annotation for their data."*

--- I agree with the authors. However, that is not what is currently being offered by the METASPACE-ML model. Most users will be looking to apply the currently available model described in this manuscript to their own data, similar to how the MSM approach has been used. How can they assess whether they receive reliable and relevant results when they do that?

Response 10:

We agree that it's imperative for the users to assess reliable and relevant results.

We would like to clarify that we have already introduced reliability measures in addition to the evaluation metrics already reported. Nonetheless, we have additionally provided a ShinyApp to help the users align their datasets against different contexts and make an informed decision.

Action taken:

Same as Response 4

Comment 11: *"Our study represents a major advancement in metabolite annotation for spatial metabolomics by introducing the first machine learning model trained and evaluated on a large collection of diverse imaging MS datasets that consistently outperforms the rule-based approach."*

--- The manuscript reports the diversity of the training data that was used and, as the manuscript indicates, the coverage of the space of potential MSI experiments is rather limited. For example, TOF analyzers, probably one of the most common mass analyzer types for MSI experiments, is not (well) represented in the training data. METASPACE-ML seems quite good in the MSI data types on which it is trained, but given that it has (nearly) only seen Orbitrap and FTICR training examples, it does not seem suited for use on TOF-based MSI datasets. Does this not narrow the scope of applicability for METASPACE-ML? This point is e.g. relevant to METASPACE-ML's application to lipid annotation, since it is quite common for lipid MSI datasets to be acquired on a TOF instrument.

If the argument is that spatial metabolomics is done mostly with Orbitraps and FTICRs (I'm not sure that statement would hold, but let us assume), then the argument only further underlines the narrow scope of where METASPACE-ML can be reliably applied.

Please treat the point of training and applicability of the model more prominently in the manuscript.

Response 11:

We have extended the discussion by including a paragraph about data from QTOF-based instruments. We assume that the reviewer referred to QTOF and not regular TOF because regular TOF is not commonly used for spatial metabolomics due to the limited resolving power and mass accuracy. Specifically, we acknowledge that our model is not trained on QTOF datasets because

QTOF imaging mass spectrometers are still less widespread for spatial metabolomics compared to Orbitrap and FTICR-based instruments. We would like to emphasize that once such datasets will be available in abundance, a new model can be easily trained with the proposed framework of the features, evaluation, and ML-based based score directly applicable to such data.

Action taken:

Same as Response 5, moreover we have added a passage on this topic to the Discussion.

Comment 12: *"we considered technology-biology configurations or contexts relevant for practitioners in the imaging mass spectrometry field"*

--- What constitutes a technology-biology configuration? Does this mean a combination of the six properties available for each dataset? Can it be a more broad combination of only five of the six properties? Please be precise in the definition of a context. Also, please specify how a practitioner in the mass spectrometry imaging field should use these contexts? At what point is the user's dataset a match for a specific context? When all six properties match? When four of the six properties match? Please elaborate.

Response 12:

We agree with the reviewer and apologise for the confusion.

The context is defined by the combination of all six metadata variables for each dataset. Some variables are specific to the technology used (such as source, analyzer, and m/z range), while others are specific to the experiment (such as sample type, organism, and polarity). We selected these six variables because they can consistently be defined.

Action taken:

We have provided a ShinyApp to help the user match those variables based on their dataset to the defined context in the manuscript. Using this app, the user can get more information about which datasets the model was trained and tested on, as well as being able to evaluate the performance of the model on those datasets.

We also further elaborated on the definition of a context in the corresponding methods section and added a paragraph explaining in more details how the ShinyApp can be used.

Comment 13: *Figure 2C and 2D*

--- Why is Figure 2C's training data missing the "mz_class" dataset parameter while Figure 2D's test data includes it?

Response 13:

We apologise for the discrepancy.

Action taken:

Figure 2C was changed to include the "mz_class" to match Figure 2D.

Comment 14: *"the balanced representation of MALDI vs. DESI-based human datasets in our evaluation set enables a fair comparison"*

--- *What do the authors mean by fair here? Figures 2C and 2D show MALDI dominating the DESI representation, also in the human subset. Please explain how to get to this statement on fairness.*

Response 14:

We agree with the reviewer and apologise for the oversight.

However, we would like to clarify that the dominance of MALDI datasets over DESI datasets in these figures results from the inherent imbalances in the METASPACE knowledge base, driven by community submissions. Despite this, we believe that our approach allows for an unbiased evaluation of METASPACE-ML's performance within each specific context, ensuring that the model's capabilities are assessed fairly within each context.

Action taken:

We understand that the term "balanced" might cause confusion when comparing the overall distribution of datasets. Therefore, we have removed all instances of the word "balanced" that could be misinterpreted as "representative" or "comparable." In the discussion, we emphasised how fixing the context size can facilitate comparative assessment without using the term "balanced."

Comment 15: *"we believe that this contextualization allows for an unbiased evaluation of METASPACE-ML's performance within each specific context, without compromising its assessment in other contexts."*

--- *If the contexts only play a role in the evaluation of the model post-training, how do they affect the issue of coverage, applicability, and balance of the trained model?*

The sampling procedure explained in the Methods section of the manuscript increases the coverage within the datatypes already used for training, but it does not increase the coverage towards more data types than the ones already considered.

Furthermore, even if performance within a context is high quality, how does one find out that their own dataset is adequately covered by the datasets captured by that context?

"We have now provided a more detailed assessment of the evaluation results per context which stratifies the results based on polarity, organism, sample-type, analyzer, ionization source, and molecular species (Lipids, small-molecules, both)."

--- *Same question as for the previous remark.*

"we have evaluated reliability of the model per context to ensure fair and unbiased comparisons."

--- *Same question as for the previous remark.*

Response 15:

We agree with the reviewer regarding the imbalance and have changed the text accordingly.

Regarding applicability, we would like to clarify that the model can be applied to a newly submitted dataset without considering context and we already show that the model performs well on contexts that haven't been seen by the model.

That's because the objective function and the input to the model is independent from the defined context, rather concerns the ion's feature scores primarily. On the other hand, when it comes to applicability, contexts help the user align their dataset against similar datasets, and providing stratified evaluation per context, helps the user make an informed decision knowing that the model performs well or not so well on similar datasets.

To measure coverage, we can provide a straightforward score by calculating the percentage of coverage for an unseen dataset. This is done by comparing the number of annotations previously seen by the model with the total number of annotations captured.

The challenging aspect is defining "adequate". We believe that it should be defined based on performance which is dictated by evaluation metrics, rather than how many datasets have seen such annotations. Context only interferes with selection of datasets and evaluation, but not in building the decision trees based on which the prediction score is calculated. Therefore, relying solely on training datasets to determine adequate coverage won't be beneficial for the user. It shouldn't influence the user's assessment of the model's performance. Instead, users should base their judgments on evaluations conducted on unseen datasets that are similar to their own.

Action taken:

- Same as response 14 regarding using the word "balanced"
- Same as response 12 regarding matching datasets to context
- Added % coverage for testing datasets in supplementary table S5.

Comment 16: *"In summary, the balanced and context-specific selection of datasets helped us select a large representative set of datasets and will provide the end-users the granularity of the represented sample types and protocols to judge on the reliability of the model predictions."*

--- This statement suggests that the responsibility for assessing whether prediction results are reliable for a user-provided dataset lies with the user. On the basis of what information can the user assess this? First, at what point should a user consider their dataset as being matched by a context? Please provide guidance. Second, even if a user's dataset can be assigned as being a member of a particular context, that does not mean that the training datasets employed in that context are relevant or representative for the content of the user's dataset. It is very good that the authors are adding datasets compared to a previous iteration of this manuscript. However, this does not address the issue of whether METASPACE-ML predictions are reliable for a user's unseen dataset.

Response 16:

We would like to clarify that, when it comes to reliability, we have provided multiple evaluation metrics including a reliability score for each dataset that can inform the users about the reliability of the model predictions for a given dataset irrespective of context.

Regarding contexts, we provide evaluation results for testing datasets (unseen by the model) that can be matched to a new dataset using our definition of context as explained in Response 12.

Once matched, the user should be able to assess how well the model performs on similar datasets. We acknowledge that there is a possibility that their dataset might differ significantly from those in the matched context. Since we use only core metadata to define context, we cannot guarantee that the model will perform equally well on their dataset as it did on similar ones. We can only provide transparent and detailed assessment on selected similar datasets which represent how well the model performs in a given context.

Action taken:

Same as Response 12.

Comment 17: *"each with their unique combination of contexts encompassing both analytical and biological metadata to provide a more granular approach for end-users to align and cross-reference their datasets with the training and evaluation datasets"*

--- How should users align and cross-reference their datasets with the datasets used to build the model? If the argument is that users should see if one of the contexts examined in the paper is similar to the context of their own data, there are still two issues preventing a user from being able to make that assessment:

1) At what point is a training context sufficiently close to the user's dataset? What if different contexts could potentially fit the user's dataset and the performance for these contexts is quite different? Is it sufficient to match on the analyzer, or does the user's dataset need to match several meta-parameters? How is the user equipped to make this assessment? The experts on the model, i.e. the authors, seem better equipped to make that call?

2) Even if a user's dataset can be correctly aligned with a specific trained context, how can the user assess whether that context has seen examples relevant to the dataset of the user (enabling good prediction)? Stratifying the dataset database into specific contexts does not solve the fundamental problem of whether the model has seen examples relevant to the new dataset, i.e. whether the user's dataset fits the trained distribution. It is not clear from the manuscript how the user can assess whether the results are reliable for their own dataset.

The argument in the manuscript seems to be that the model has been trained on the largest collection of MSI datasets available. However, that is not the same as the model being trained on a collection of MSI datasets that is sufficiently large, and the latter is necessary to ensure reliability in general, a claim suggested by the title, abstract, and introduction of the manuscript. If the training data is not sufficiently large, as seems to be the case here, the alternative is to assess whether a dataset one wants to predict for matches the training data sufficiently, but this does not seem to be provided here.

Response 17:

We respectfully disagree with the reviewer's claim that the training data is not sufficiently large, as this assertion doesn't clarify what is meant by "sufficient.". Nevertheless, the solution outlined in Response 15 should help determine whether a dataset sufficiently matches the training data to a certain extent.

In Response 15, we proposed a method to assess coverage based on annotations previously seen by the model. However, it is important to note that the model learns collectively from the feature scores of multiple annotations, which forms the basis for its prediction score. Therefore, while we

can provide coverage metrics, the model's performance does not depend solely on the number of annotations it has encountered during training. We would like to emphasise that the model should be judged primarily by its performance on unseen datasets, using the provided evaluation metrics, as this is the main criterion for assessing the model's reliability.

Action taken:

Same as Response 15.

Comment 18: *"The datasets were classified based on possible combinations of the following 6 variables : polarity, ionization source, mass analyzer, mz class, sample type and species."*

--- Which contexts were considered exactly? All possible combinations of these six variables?

Response 18:

We apologise for the oversight

Action taken:

We modified Table S5 by adding column context and changed the methods accordingly to further explain which contexts are selected.

Comment 19: *"for some models, certain contexts are not represented due to insufficiency of available datasets to cover both testing and training."*

--- Which models saw which contexts? Please specify.

Response 19:

Same as Response 18

Comment 20: *"final list of selected datasets for each context and kingdom can be found in Supplementary Table 5."*

--- The kingdom is indicated in Table S5, but I don't seem to be able to find an easy mapping to a context. Which contexts (or parameter combinations) were actually trained? Are these all possible combinations of the 6 variables, or less? If I missed it, please point me to a definitive list of the contexts. If not, please provide a list and their connection to the models.

Response 20:

We apologise for the oversight.

Action taken:

Same as Response 18.

Comment 21: *"while maintaining the relative size of each context constant."*

--- Please elaborate. Two independent CatBoost models were built for animal and plant datasets independently, but which contexts play a role in the training phase is not very clear. Please provide an exact record of the training of each of the two models. Please provide an exhaustive list of the contexts used. Since the contexts are central to the improvements made to the previous version of this manuscript, it is important to give a strong and rigorous treatment of this aspect.

Response 21:

We apologise for the oversight.

Action taken:

Same as Response 18.

Comment 22: Figure S1 and S2

--- Please provide a clear definition and enumeration of what constitutes a "context". Figure S1 mentions six contexts: polarity, source, analyzer, mz_class, sample_type, and species. Figure S2, however, mentions that there are 24 contexts in certain situations (along the vertical axis labeled "number of contexts"). In Figure S4, it is mentioned that "A flow from the first to the last node represents a single context", but in that figure "mz_class" is not part of the Sankey diagram, so does that mean that a context is not specific to "mz_class"? That seems in contradiction with Figure S2. Also, the drawing in Figure S4 suggests that there are 32 different contexts, which does not line up with statements elsewhere. Then again, Figure S8, shows 36 contexts based on the labels polarity, species, sample_type, but it takes source+analyzer together as one and at the same time seems to leave mz_class out of the specification. Also in Figure S8, 13 of the 36 contexts seem to be empty, presumably due to not enough datasets in those label combinations? Does this mean there are actually 23 non-zero dataset contexts? If so, does that not mean the training data is still unbalanced? The matter is confused further by Figure 4C in the main manuscript, suggesting there are 24 contexts in total in the animal-based testing datasets, and Figure S10 suggests there are 7 contexts in total in the plant-based testing datasets. While one could consider Figure 4C to line up with Figure S1, their total context number differs from Figure S11, which lines up with S8 if we deduct the empty boxes. To avoid these ambiguities, please be precise in definitions and rigorous in wording throughout.

Furthermore, if the user is to assess whether METASPACE-ML is reliable for their own dataset and there's only one model for several contexts, does the unbalanced distribution of dataset numbers across different contexts not mean that the model would be skewed towards the majority meta-information labels in the total training data? How does a user know whether their own dataset fits sufficiently with the majority representatives that the final model was trained on?

The manuscript can use further elaboration on what constitutes a context exactly, what is an exhaustive list of the contexts considered, and how is the interaction between contexts and any other steps performed, e.g. cross-validation?

Response 22:

We agree with the reviewer that contexts should be better defined and apologize for the confusion.

We have better defined contexts in the text. In this manuscript a context is defined by all 6 variables (polarity, source, analyzer, mz_class, sample_type, and species). Accordingly we would like to clarify the following regarding the mentioned figures :

- Figure S1 : Shows a breakdown of plant datasets in METASPACE broken down by the above 6 variables used to define a context. This represents the pool of datasets from which training and testing datasets were selected.
- Figure S2 : Shows 5 different models where x-axis represents the number of datasets **per context** and y-axis represents how many contexts are covered by the **training** datasets. In the paper, we settled on 30 datasets per context, covering 20 contexts for animals and 6 for plants. We explain that choice in the results section (**Representative selection of datasets**).
- Figure S4: We understand that the figure can be potentially misinterpreted due to its static nature, making it difficult to explore the flows in more detail as it shows mostly relative size of each variable. For example, only datasets acquired in positive polarity are coupled with Orbitrap, not negative. And the figure might not reflect that. We have added the exact contexts used in supplementary tables, S2 and S5 and provided an interactive visualization in an accompanying ShinyApp to better communicate which contexts are involved.
- Figure S8 : There are actually 24 contexts available for **testing** datasets. We have added those contexts in Supplementary table S5. adjusted the caption accordingly to reflect this and fixed the plot to highlight the missing context.
- Figure 4C will now align with Figure S8 as there are 24 contexts in animal-based testing datasets.

Action taken:

- The mentioned supplementary figures and their corresponding main have been changed to add mz_class in all figures where it was missing.
- Added exact contexts used in Supplementary tables S2 and S5.
- Main text has been rephrased accordingly as mentioned in Response 12 to better define context.
- Solutions proposed in Response 15 should provide guidance in matching a dataset to a context and evaluating sufficient coverage.
- The ShinyApp mentioned in the above replies should help better define context and to explore the results interactively which avoids the misinterpretation of Sankey plots.

Other remarks

Comment 23: *My original comment was "When a dataset's m/z range is labeled as 'high', does this mean that its m/z range goes higher than m/z 500? How high? What is the maximum m/z measured? Please report a summary of m/z ranges covered." In response, a new sentence in the manuscript reads as follows "Finally, datasets were also classified based on their mz range, where datasets having maximum mz <= 400 are classified as small molecules and those having minimum mz > 500 are classified as Lipids. The rest of datasets with min mz < 500 and max mz > 400 are classified as "Lipids and small molecules""*

--- This change does not answer my original question. Please specify the maximum m/z of the ranges. For example, for the "min mz < 500" datasets, what is the extent (in m/z units or Th) of

their ranges covered? Up to what m/z value do we consider molecular species in METASPACE-ML, and why is that an appropriate range?

Response 23:

We apologise for the misunderstanding.

Action taken:

We have modified Table S2 and added information about min and max m/z for each dataset.

Comment 24: *"we have validated some of the newly captured annotations by METASPACE-ML compared to the rule-based approach by performing LC-MS/MS bulk analysis of mouse brain datasets from our previous paper (ref)"*

--- Which paper is being referred to here?

Response 24:

We were referring to our original paper entitled "FDR-controlled metabolite annotation for high-resolution imaging mass spectrometry"

<https://www.nature.com/articles/nmeth.4072>

Comment 25: *"The mean average precision (MAP) was used to evaluate the quality of target vs decoy ion rankings."*

--- Many different measures could be used to evaluate performance here. Each measure has its own sensitivities and blind spots. Please elaborate why MAP is the right measure to use here.

Response 25:

We thank the reviewer for their comment.

We chose MAP because it's particularly effective in evaluating the precision of the top-ranked ions. MAP considers the precision at each rank position and averages these values, giving a more nuanced view of the ranking quality. This makes it a robust metric for our purpose, where both the order and the presence of target ions in the top ranks significantly impact the model's performance when it comes to quality of ranking.

Action taken:

We added a reference in methods to support why MAP is the right measure to rely on when it comes to ranking.

Comment 26: *"A dot represents a dataset and y-axis represents the Log10 absolute difference where negative values are those where MSM had more annotations than METASPACE-ML and vice-versa."*

--- There are quite some dots on the negative side, suggesting that for those datasets MSM finds certain annotations that are not found by METASPACE-ML. Please elaborate more on the nature of those annotations.

Response 26:

We thank the reviewer for pointing this out.

Annotations captured by MSM but not by METASPACE-ML at low FDR (e.g. 10%) are mainly due to MSM's sensitivity to its feature scores, particularly the spatial score. In certain datasets, the target ion may have a high spatial score while decoys have low spatial scores, resulting in a high MSM rank and low FDR for the target ion. Conversely, METASPACE-ML prioritizes spectral scores over spatial scores, which can cause higher FDR for these targets as the decoy scores are closer to the target ion scores.

Action taken:

We have added the above explanation in the discussion for further clarification.

Comment 27: "our approach has the potential to enhance imaging mass spectrometry and spatial metabolomics, and to have far-reaching implications for biology, medicine, and pharmacology."

--- Please avoid these rather broad statements and instead provide the exact areas within which reliability of the model is high.

Response 27:

We agree with the reviewer.

Action taken:

We have removed the statement from the manuscript.

Comment 28: "based on their m/z range"

--- Please use "m/z" in the text of the manuscript. "mz" is incorrect.

Response 28:

We agree with the reviewer and apologise for the oversight.

Action taken:

We have rephrased the text accordingly by replacing all instances of mz to m/z.

Comment 29: "30 testing datasets per context were selected first for each kingdom"

--- Why 30? Why is this number a reasonable value to use?

Response 29:

We thank the reviewer for pointing this out.

The selection of 30 datasets per context was chosen to strike a balance where including more datasets could lead to less diversity of contexts, whereas including fewer might risk compromising statistical reliability due to a smaller sample size potentially leading to less robust evaluation.

Action taken:

We have added a paragraph in the methods to explain why we chose 30 datasets.

Comment 30: *Equation 3 seems to imply that the difference between the theoretical m/z value of the first isotope and the observed mean m/z value of the first isotope can never exceed 1. Is that the case? Is this enforced beforehand? If so, I don't think I saw it in the preceding method description. If not, what happens if the m/z difference is larger than one?*

Equation 4

--- Same questions as for Eq. 3. in the previous remark.

Response 30:

We thank the reviewer for their comment.

This was changed compared to the previous paper to make sure that all the features provided to the model are in the scale of 0-1 where higher scores are favourable. Therefore, we scaled the m/z error accordingly so that scores closer to 1 corresponds to lower difference / error. For all annotations, the m/z difference is never larger than one so this equation holds.

Comment 31: *"we choose beta = 0.5 to put more weight on the precision"*

--- *Why is 0.5 an appropriate parameter value for beta?*

Response 31:

We thank the reviewer for their comment.

By setting beta to 0.5, we ensure that the introduced reliability score reflects the importance of minimizing false positives, which are more critical than false negatives for our model predictions. Moreover, the choice of beta = 0.5 allows us to strike a balance that emphasises precision without completely disregarding recall in our assessment of reliability of FDR thresholds.

Typos

- "mz <= 400"
--- The property mentioned here is the mass-to-charge ratio and should therefore be written as "m/z". The authors use "mz", which (incorrectly) implies mass multiplied with charge to be the property. Please be exact in notation.
- "we can calculate the proportion of overlapping intervals after shifting m/z values by a given ppm"

--- Equation (1) does not seem to be shifting the positions of m_i and m_{i+1} , but rather checking the distance.

- Table S5

--- This table seems to use "Lipids_and_small_moleclues" throughout. Please correct the spelling.

We thank the reviewer for the comprehensive review and have corrected the mentioned typos.

Responding to Reviewer #3

The authors adequately addressed my comments.

We thank the reviewer for their review.

Reviewer #1 (Remarks to the Author):

In revised version of the manuscript authors addressed all the critique points

Reviewer #2 (Remarks to the Author):

In previous review rounds, two main issues were highlighted regarding the METASPACE-ML model and the submitted manuscript:

- (1) Limited coverage of the spatial metabolomics space in the training data.
- (2) Contexts and reliability of METASPACE-ML predictions.

I appreciate the efforts conducted by the authors to address these matters, as well as the development of a ShinyApp to help the user of METASPACE decide some of the alignment issues for their own datasets. In the current version of the manuscript, the authors seem to have revised the text primarily in terms of stepping away from concepts such as balanced and general representation of the whole metabolomics space (e.g. removing such words from the manuscript text), instead acknowledging that the model can have applicability in particular subareas of the metabolomics and instrument space (e.g. focusing on the contexts concept they introduced). The acknowledgement of the narrower applicability of the METASPACE-ML model is a reasonable manner to address this matter.

However, it should be noted that the original comments therefore still stand and are not really resolved. Also, with the addition of the ShinyApp, the assessment of whether a new dataset supplied by a user is sufficiently close to a context that was trained on still seems to be passed on to the user, who is (even with the app) not particularly well-equipped to make that call.

While I appreciate the acknowledgement of METASPACE-ML's narrower applicability on certain subsets of the metabolomics space (i.e. certain analyzer types, certain tissue types, etc.) in, for example, the Results section with subsection "Representative selection of datasets", it does not seem like the Abstract, Introduction and, more importantly, the title reflect that narrower applicability. For example, the abstract still states "various spatial metabolomics contexts", implying broad coverage rather than conveying applicability in select subareas of the metabolomics space.

In the latest rebuttal document, the authors state that they "rephrased the text in abstract and introduction to better describe the scope of METASPACE-ML." --- I'm not convinced this is clear from their current formulation. The title, abstract, and introduction still communicate a broader scope than can be delivered by the model and the manuscript.

Since the current manuscript seems to embrace the narrower applicability of the model, at least the title should reflect that narrower focus. The current title still reflects a broader applicability than is offered by the content of the manuscript and the model. Please make sure the title reflects the limitations in coverage and applicability. One suggestion would be to add something like "in specific contexts" to the title to make that point more clear.

Summary of changes

We have introduced the following modification to the main text:

- Added further clarifications on how the Shiny App should be used in the corresponding methods section
- Rephrased text in abstract and introduction to better communicate context-specificity.
- Changed the manuscript title to “METASPACE-ML: Context-specific metabolite annotation for imaging mass spectrometry using machine learning”

Responding to Reviewer # 2

Comment 1: *Also, with the addition of the ShinyApp, the assessment of whether a new dataset supplied by a user is sufficiently close to a context that was trained on still seems to be passed on to the user, who is (even with the app) not particularly well-equipped to make that call.*

Response 1:

We acknowledge that the app does not specify which context users should rely on, leaving it up to them to decide whether METASPACE-ML is appropriate for their dataset. However, the app is designed to enhance transparency regarding model performance and to allow users to explore datasets and the model's outcomes within specific contexts. These contexts are based on variables that are likely present in every dataset and experimental design, ensuring that users can provide this minimal information. With context-specific results available, users can better evaluate whether METASPACE-ML outperforms its predecessor. Future enhancements in integrating the ML model with METASPACE and the ShinyApp will further assist users in assessing the reliability of annotation results.

Action taken:

We have added further clarification and disclaimer on how the app should be used in the corresponding methods section.

Comment 2: *While I appreciate the acknowledgement of METASPACE-ML's narrower applicability on certain subsets of the metabolomics space (i.e. certain analyzer types, certain tissue types, etc.) in, for example, the Results section with subsection "Representative selection of datasets", it does not seem like the Abstract, Introduction and, more importantly, the title reflect that narrower applicability. For example, the abstract still states "various spatial metabolomics contexts", implying broad coverage rather than conveying applicability in select subareas of the metabolics space.*

In the latest rebuttal document, the authors state that they "rephrased the text in abstract and introduction to better describe the scope of METASPACE-ML." --- I'm not convinced this is clear from their current formulation. The title, abstract, and introduction still communicate a broader scope than can be delivered by the model and the manuscript.

Response 2:

We apologize if the modifications were unclear. We have further narrowed the scope in the abstract and introduction .

Action taken:

We have rephrased the text in abstract and introduction to better communicate context-specificity over broader coverage.

Comment 3: *Since the current manuscript seems to embrace the narrower applicability of the model, at least the title should reflect that narrower focus. The current title still reflects a broader applicability than is offered by the content of the manuscript and the model. Please make sure the title reflects the limitations in coverage and applicability. One suggestion would be to add something like "in specific contexts" to the title to make that point more clear.*

Response 3:

We thank the reviewer for the feedback. We have changed the title accordingly to highlight context-specificity.

Action taken:

The manuscript title was changed to "METASPACE-ML: Context-specific metabolite annotation for imaging mass spectrometry using machine learning"